

# High-spatial-resolution monthly temperature and precipitation dataset for China for 1901–2017

Shouzhang Peng[1], Yongxia Ding[2], Zhi Li[3]

[1] State Key Laboratory of Soil Erosion and Dryland Farming on the Loess Plateau, Northwest A&F University, Yangling, 712100, China
[2] School of Geography and Tourism, Shaanxi Normal University, Xi'an, 710169, China
[3] College of Natural Resources and Environment, Northwest A&F University, Yangling, 712100, China

*Correspondence to*: Zhi Li (lizhibox@nwafu.edu.cn  szp@nwafu.edu.cn)

**Abstract:** High-spatial-resolution and long-term climate data are highly desirable for understanding climate-related natural processes. China covers a large area with a low density of weather stations in some regions, especially mountainous regions. This study describes a high-spatial-resolution (0.5', ~1 km) dataset of monthly temperatures (minimum, maximum, and mean TMPs) and precipitation (PRE) for the main land area of China for the period 1901–2017. The dataset was spatially downscaled from raw 30' climatic research unit (CRU) time series data and validated using data from 745 weather stations across China. Compared to raw CRU data of low spatial resolution, the mean absolute error decreased by 0.56 °C for the TMPs and 10.1 % for PRE, the root-mean-square error decreased by 0.65 °C for the TMPs and 11.6 % for PRE, and the Nash–Sutcliffe efficiency coefficients increased from 0.83 to 0.95 for the TMPs and from 0.63 to 0.76 for PRE. Indirect validations from site-scale observations indicated that the dataset captured the climatology well, as well as the annual and seasonal monotonic trends in each climatic variable considered. We concluded that the new high-spatial-resolution dataset is sufficiently reliable for use in investigation of climate change across China. This dataset will be useful in investigations related to climate change across China. The dataset presented in this article is published in the Network Common Data Form (NetCDF) at http://doi.org/10.5281/zenodo.3114194 for precipitation (Peng, 2019a) and http://doi.org/10.5281/zenodo.3185722 for temperatures (Peng, 2019b). The dataset includes 156 NetCDF files compressed with zip format and one user guidance text file.

## 1 Introduction

High-spatial-resolution and long-term climate data are required for accurate investigations of changes in climate and in climate-related phenomena that affect hydrology, vegetation cover, and crop production (Gao et al., 2018; Caillouet et al., 2019; Peng et al., 2018; Peng and Li, 2018). Although meteorological observation networks incorporate increasingly greater numbers of weather stations and the contributions of increasing numbers of governments and researchers around the world, observation networks still suffer from low station density and spatial resolution (Caillouet et al., 2019; Peng et al., 2014), especially in mountainous areas (Gao et al., 2018) because the installation and maintenance of weather stations in such areas



are challenging (Rolland, 2003). Accordingly, several interpolation methods, such as inverse distance weighting, kriging methods, and regression analysis are usually used to generate meteorological data for those ungauged areas (Li et al., 2010; Li et al., 2012; Zhao et al., 2004; Atta-ur-Rahman and Dawood, 2017; Peng et al., 2014). However, the accuracy of the results of these interpolation methods depends on the station density (Gao et al., 2018; Peng et al., 2014). Therefore, it is

necessary to use climatic proxy data to generate long-term and high-spatial-resolution climate data.

Proxy monthly temperature (TMP) and precipitation (PRE) data products are released by several climate research organizations, such as the General Circulation Models (GCMs) of Intergovernmental Panel on Climate Change (Brekke et al., 2013), the Climatic Research Unit (CRU) of the University of East Anglia (Harris et al., 2014), the Global Precipitation Climatology Centre (GPCC) (Becker et al., 2013), and Willmott & Matsuura (W&M) (Matsuura and Willmott, 2015a, b).

These products have long time series (> 100 years) and moderate spatial resolution (≥ 30'). Compared with GCMs products, CRU, GPCC, and W&M products are generated from data from observational stations and thus have higher reliability. Furthermore, compared with GPCC and W&M products, CRU products include several TMP and PRE variables, such as monthly mean TMP, maximum TMP, minimum TMP, and PRE. Therefore, CRU products have been widely employed to investigate climate effects around the world (Kannenberg et al., 2019; Lewkowicz and Way, 2019; Bellprat et al., 2019).

Although CRU products offer the advantage of reflecting long-term climate effects, their spatial resolution (30', approximately 55 km) limits their ability to reflect the effects of complex topographies, land surface characteristics, and other processes on climate systems (Xu et al., 2017; Peng et al., 2018). This also prevents CRU data from providing realistic and reliable climate change information on fine scales, which is imperative when developing adaptation and mitigation strategies that are suitable for use on local scales (Giorgi et al., 2009; Peng et al., 2019). Therefore, it is necessary to spatially

downscale and correct CRU climate data.

Previous studies have shown that the Delta downscaling framework performs well in downscaling climate data (Mosier et al., 2014; Peng et al., 2018; Peng et al., 2017; Wang and Chen, 2014; Brekke et al., 2013). This framework uses a low-spatial-resolution monthly time series and high-spatial-resolution reference climatology as inputs. The high-spatial-resolution climatology must be physically representative, fine-scale distribution of meteorological variables over the

landscape of interest (Mosier et al., 2014; Peng et al., 2017). As a result of the incorporation of a high-spatial-resolution reference climatology, downscaled results often have higher accuracy than raw data in comparison to weather station data, especially for monthly mean TMP and PRE (Peng et al., 2018). Thus, the Delta downscaling framework is able to downscale and correct low-resolution climate data.

China covers a large land area and has many mountainous areas. Even the establishment of additional weather stations has

not made it possible to fully satisfy requirements for long-term, high-spatial-resolution climate data, especially at finer geographical scales and for mountainous areas. Furthermore, the weather stations in China were established after 1950, and thus longer-term observational climate data are unavailable (Peng et al., 2018). These above shortcomings limit the types of studies that can be conducted on long-term climate change and climate change effects at fine geographical scales across China.





The objective of this study was the generation of a long-term climate dataset of high spatial resolution for the land area of China by means of downscaling CRU time series data. The specific climate data types generated included monthly TMPs (mean, maximum, and minimum TMPs) and PRE with a spatial resolution of 0.5' (approximately 1 km), from January 1901 to December 2017. After the downscaling, the raw and downscaled TMPs and PRE data were compared with observed data

from weather stations across China and were compared with three other downscaled datasets with spatial resolutions of 10', 5', and 2.5'. This latter set of comparisons was conducted to demonstrate the advantages of the downscaling framework and the accuracy of a downscaled dataset with a spatial resolution of 0.5'. Trends in the annual and seasonal TMPs and PRE across China were also investigated using the raw data, the downscaled data, and the weather station observations.

**2 Data**

Monthly mean, maximum, and minimum TMPs, as well as PREs, with a spatial resolution of 30' were obtained from the CRU TS v. 4.02 dataset (http://www.cru.uea.ac.uk) (Harris et al., 2014). This dataset covers the period from January 1901 to December 2017. To downscale this dataset to higher spatial resolutions, we obtained four high-resolution reference data sets at spatial resolutions of 10', 5', 2.5', and 0.5' from the WorldClim v. 2.0 (http://worldclim.org) (Fick and Hijmans, 2017). The reference datasets contained monthly averages of climate parameters for the period 1970–2000, generated from data

from between 9, 000 and 60,000 weather stations around the world, using the thin-plate splines interpolation method. The interpolation considered covariation with elevation, distance to the nearest coast, and three satellite-derived covariates: the maximum and minimum land surface temperature and cloud cover, obtained from the MODIS satellite platform. Thus, these reference data reflect orographic effects and observed climate information for each month.

To evaluate the performance of the downscaling procedure, observed monthly TMPs and PRE across China were obtained

from the National Meteorological Information Center of China (http://data.cma.cn/en). This dataset includes observations from 745 national weather stations (Fig. 1) for the period 1951–2016.

**3 Methods**

**3.1 Spatial downscaling**

Delta downscaling was employed to generate monthly TMPs and PRE for the period 1901–2017 at spatial resolutions of 10',

5', 2.5', and 0.5'. Figure 2 illustrates the components and steps of the Delta downscaling for PRE using the CRU 30' time series and WorldClim 0.5' climatology datasets (Peng et al., 2018). The first step (Fig. 2a) involves constructing a CRU 30' climatology for each month in the period 1970–2000. The second step involves calculating the 30' anomaly for a specific month relative to the long-term average for that month (Fig. 2b). The PRE anomaly was calculated as the ratio of the PRE in a specific month to the long-term averaged PRE for that month. The TMP anomaly was calculated as the difference between

the TMP in a specific month and the long-term averaged TMP for that month. The third step involved spatial interpolation of

the 30' anomaly to the 0.5' WorldClim grid (Fig. 2c). The final step (Fig. 2d) involved transformation of the 0.5' anomaly to the 0.5' PRE for that month using the WorldClim climatology for the corresponding month. This transformation undid the creation of the anomaly; therefore, multiplication was used for PRE, while addition was used for TMP. It should be noted that the interpolation of the anomaly can be carried out by many methods. We compared the performance of the bicubic interpolation, bilinear interpolation, and nearest-neighbor interpolation methods.

## 3.2 Evaluation of performance of downscaling procedure

Direct and indirect evaluations were conducted to assess the performance of the spatial downscaling procedure. The direct evaluation involved comparison of some statistical parameters of climatic variables associated with the raw and downscaled data with observations from weather stations. The parameters included the mean absolute error (MAE), root mean square error (RMSE), and Nash–Sutcliffe efficiency coefficient (NSE). The indirect evaluation involved comparing the climatology and trends in climatic variables associated with the raw and downscaled data with observations from weather stations.

## 4 Results

### 4.1 Direct evaluation by comparing statistical parameters

Table 1 presents the statistical parameters for the raw/downscaled and observed values for the period 1951–2016 for the 745 weather stations. The results show that (1) the downscaled data had lower MAE and RMSE values and higher NSE values than the raw data; (2) the increased spatial resolution of the WorldClim reference dataset from 10' to 0.5' resulted in decreased MAE and RMSE values and increased NSE values; (3) of the three interpolation methods employed in the Delta downscaling, the data downscaled using the bilinear interpolation method had the lowest MAE and RMSE values and highest NSE values at each spatial resolution; and (4) the performance of the Delta downscaling was better for TMPs than PRE.

Specifically, the MAE of the downscaled minimum TMP at 0.5' under bilinear interpolation decreased from 1.62 to 1.14 °C, the RMSE decreased from 1.89 to 1.37 °C, and the NSE increased from 0.89 to 0.96. For the mean TMP, the MAE of the downscaled data at 0.5' under bilinear interpolation decreased from 1.38 to 0.84 °C, the RMSE decreased from 1.54 to 1.01 °C, and the NSE increased from 0.87 to 0.97. For the maximum TMP, the MAE of the downscaled data at 0.5' under bilinear interpolation decreased from 1.88 to 1.22 °C, the RMSE decreased from 2.36 to 1.45 °C, and the NSE increased from 0.74 to 0.92. For PRE, the MAE of the downscaled data at 0.5' under bilinear interpolation decreased by 10 %, the RMSE decreased by 12 %, and the NSE increased by 21 %. Overall, the downscaled TMP and PRE data had higher accuracy than the raw CRU data.



## 4.2 Indirect evaluation by climatology comparison in terms of climatic variables

Low weather station density and interpolation of the data from these stations may introduce uncertainties in representing the climatology of China as a whole. We therefore investigated the climatology using the downscaled TMP and PRE data for 1901 to 2017. The  mean annual temperature and total precipitation were used to represent the climatology in terms of mean

TMP and PRE, while the 1 % and 99 % quantiles (Q1 and Q99) of the monthly minimum and maximum TMPs, respectively, were selected to represent the climatology in terms of minimum and maximum TMPs, because quantile temperatures are more reliable than absolute minimum and maximum TMPs if an outlier exists (Fig. 3). The Q1 of the minimum TMP for China as a whole ranged from -50.15 °C to 17.21 °C, with an average of -17.1 °C, and the lowest value corresponded to a location in the western part of the Qinghai–Tibet Plateau (Fig. 3a). The Q99 of the maximum TMP ranged from -16.33 °C to

42.27 °C, with an average of 26.88 °C, and the highest value corresponded to a location in the Turpan Basin (Fig. 3b). The annual mean TMP ranged from -34.41 °C to 26.39 °C, with an average of 6.18 °C, and the lowest and highest values correspond to locations in the western part of the Qinghai–Tibet Plateau and Hainan Island, respectively (Fig. 3c). The mean annual total PRE ranged from 3.2 mm to 4854.0 mm, with an average of 564.4 mm, and the minimum and maximum values correspond to locations in the northwestern part of the Qinghai–Tibet Plateau, near the Tarim Basin, and Taiwan Island,

respectively (Fig. 3d). The climatology for the three TMPs varies with the topography and notably decreases with orographic uplift, and the climatology for PRE decreases from the southeastern coastal region to the northwestern region.

Table 2 shows the averaged climatology for the 745 weather stations as calculated from the raw, downscaled, and observed data. The comparison results indicate that (1) the averaged climatology from the downscaled data is closer to that from the observed data than that from the raw data; (2) the averaged climatology from the observed data is greater than that

from the raw data and downscaled data, except for the mean TMP at a spatial resolution of 2.5'; and (3) an increase in the spatial resolution from 30' to 0.5' results in the averaged climatology gradually approaching that from the observed data. Specifically, the averaged climatology differences between the 0.5' downscaled and observed data are -0.13 °C for the monthly minimum TMP, -0.53 °C for the monthly maximum TMP, -0.04 °C for the annual mean TMP, and -5.8 mm for the annual total PRE. To further illustrate the ability of downscaled data to reflect the climatology, Figure 4 shows scatterplots

of observed and downscaled/raw climatology values for the period from 1951 to 2016 for the 745 weather stations. Although there are good fitting relationships between the raw and observed values for the climatology at the weather stations, better fitting relationships were obtained for the 0.5' downscaled dataset for each climatic variable. We therefore concluded that the 0.5' downscaled dataset best capture the climatology for the whole of China.

## 4.3 Indirect evaluation by comparing temporal trends in climatic variables

Figure 5 presents the spatial patterns in the trends in TMPs and PRE and their significance levels across China for the 0.5' downscaled data for the period from 1901 to 2017, based on linear regressions between climatic values and years in the time series. The 95% significance level was selected to represent the significance of the trend for any climatic variable. The





annual minimum TMP exhibited a significant upward trend, from 0.018 °C 10 yr⁻¹ to 0.240 °C 10 yr⁻¹, with an average of 0.131 °C 10 yr⁻¹, over areas accounting for approximately 94.17 % of the total land area of China (Figs. 5a and e). The annual maximum TMP exhibited a significant upward trend, from 0.016 °C 10 yr⁻¹ to 0.171 °C 10 yr⁻¹, with an average of 0.081 °C 10 yr⁻¹, over areas accounting for approximately 80.85 % of the total land area of China (Figs. 5b and f).

5 Meanwhile, the annual maximum TMP exhibited a significant downward trend, from 0.019 °C 10 yr⁻¹ to 0.034 °C 10 yr⁻¹, with an average of 0.027 °C 10 yr⁻¹, in areas accounting for only approximately 0.33 % of the land area of China (Figs. 5b and f). The mean annual TMP exhibited a significant upward trend, from 0.017 °C 10 yr⁻¹ to 0.189 °C 10 yr⁻¹, with an average of 0.104 °C 10 yr⁻¹, over areas accounting for approximately 90.92 % of the total land area of China (Figs. 5c and g). The annual PRE exhibited a significant upward trend, from 0.11 mm 10 yr⁻¹ to 21.206 mm 10 yr⁻¹, with an average of 3.306

10 mm 10 yr⁻¹, over areas accounting for approximately 22.02 % of the total land area of China (Figs. 5d and h). Meanwhile, the annual PRE exhibited a significant downward trend, from 0.13 mm 10 yr⁻¹ to 30.321 mm 10 yr⁻¹, with an average of 7.147 mm 10 yr⁻¹, over areas accounting for only approximately 2.01 % of China (Figs. 5d and h).

Figures 6-9 and Table 3 present the annual and seasonal trends in the raw, 0.5' downscaled, and observed TMPs and PRE for the 745 weather stations for the period from 1951 to 2016. The results show that (1) the annual and seasonal 0.5'

downscaled TMPs and PRE are closer to the observations than the raw values in the time series, and (2) the annual and seasonal trends in the 0.5' downscaled data are closer to the observed trends than the raw trends, except for the summer trend in the minimum TMP. Furthermore, the annual and seasonal trends in the 0.5' downscaled TMPs are underestimated. Specifically, for the minimum TMP, there are underestimates of 0.046 °C 10 yr⁻¹ for the annual trend, 0.02 °C 10 yr⁻¹ for the spring trend, 0.068 °C 10 yr⁻¹ for the summer trend, 0.06 °C 10 yr⁻¹ for the autumn trend, and 0.047 °C 10 yr⁻¹ for the winter

trend (Fig. 6 and Table 3). For the maximum TMP, there are underestimates of 0.047 °C 10 yr⁻¹ for the annual trend, 0.064 °C 10 yr⁻¹ for the spring trend, 0.058 °C 10 yr⁻¹ for the summer trend, 0.057 °C 10 yr⁻¹ for the autumn trend, and 0.063 °C 10 yr⁻¹ for the winter trend (Fig. 7 and Table 3). For the mean TMP, there are underestimates of 0.059 °C 10 yr⁻¹ for the annual trend, 0.063 °C 10 yr⁻¹ for the spring trend, 0.047 °C 10 yr⁻¹ for the summer trend, 0.066 °C 10 yr⁻¹ for the autumn trend, and 0.053 °C 10 yr⁻¹ for the winter trend (Fig. 8 and Table 3). For the PRE, there are overestimates of 0.487 mm 10 yr⁻

¹ for the annual trend and 0.293 mm 10 yr⁻¹ for the summer trend and underestimates of 0.594 mm 10 yr⁻¹ for the spring trend, 0.281 mm 10 yr⁻¹ for the autumn trend, and 0.492 mm 10 yr⁻¹ for the winter trend (Fig. 9 and Table 3). The differences between the 0.5' downscaled and observed trends for each climatic variable are relatively small, and thus the 0.5' downscaled data can be judged to represent the temporal variations and trends in TMPs and PRE across China. However, there are some trends that warrant particular attention. Specifically, the autumn trend in the minimum TMP is significant at

the 95% level for the observed data but not significant for the downscaled data, and the summer trend in the maximum TMP is positive for the observed data but negative for the downscaled data. Thus, the 0.5' downscaled data may not be able to capture the temporal trends for these cases.



## 5 Data availability

The 0.5' downscaled dataset developed in this study is published in network Common Data Form (NetCDF) at http://doi.org/10.5281/zenodo.3114194 for precipitation (Peng, 2019a) and http://doi.org/10.5281/zenodo.3185722 for temperatures (Peng, 2019b). The dataset includes monthly minimum temperatures, maximum temperatures, mean temperatures, and total precipitation for the period from January 1901 to December 2017. Because of the availability of raw CRU data and the spatial resolution of the reference climatology, the data covers most of the land area of China, with a geographic range of 18.2–53.5° N and 73.5–135.0° E. The total number of grids is 13,808,747. To reduce the size of the NetCDF file, the data for each climatic variable are divided into intervals of 3 years. TMPs and PRE are expressed to precisions of 0.1 °C and 0.1 mm, respectively, and they are stored using int16 format. Thus, each file contains 36 months of data and requires 2.42 GB of storage space. This file size should be convenient for processing by modern computers, and subparagraph storage in the time series can satisfy needs for quick data access for specific period. Each file name indicates the data contained in the file, in the format "data type"_"beginning year"_"ending year".nc. For example, the file named tmn_1901_1903.nc contains minimum temperature data for 1901 to 1903. The total number of NetCDF files is 156, and the disk usage of the dataset in nc format is approximately 378 GB. After compression with zip format, it is approximately 300 MB for one file and 47.8 GB for total files. This dataset will be updated yearly because updating of the CRU TS dataset occurs yearly, and new data will be available for download from the website identified above.

The raw 30' monthly TMP and PRE data from 1901 to 2017 were obtained from the CRU TS v. 4.02 dataset (http://www.cru.uea.ac.uk/data, last access: 25 Apr 2019). The high-resolution reference data at spatial resolutions of 10', 5', 2.5', and 0.5' for TMP and PRE were supported by WorldClim v. 2.0 (http://worldclim.org/version2, last access: 25 Apr 2019). The observed monthly meteorological data from the 745 weather stations across China were provided by the National Meteorological Information Center of China (http://data.cma.cn/en, last access: 25 Apr 2019).

## 6 Discussion, limitations, and recommendations

Although the raw CRU data with a 30' spatial resolution was not evaluated as being poor, the 0.5' downscaled data was evaluated as being better, with deviations decreased by approximately 30 %–40 % for the TMPs and 10 % for the PRE, relative to the raw CRU data (Table 1). Thus, corrections to the raw CRU data are needed. Many factors contribute to the deviations, such as observational errors, sample size, and operator errors in gathering the raw CRU data. However, there has been little work done to address this issue. Previous studies have indicated that topographic information (e.g., elevation, location, slope, and aspect) may be key factors in correcting deviations, especially in mountainous areas (Gao et al., 2018; Peng et al., 2014; Gao et al., 2017). Therefore, a high-resolution reference climatology containing detailed topographic information was used in this study to downscale the 30' raw CRU data to a dataset of 0.5' monthly TMPs and PRE for the period from January 1901 to December 2017 across China, which has a low density of weather stations in mountainous areas



and few observations obtained before 1950. To the best of our knowledge, this 0.5' downscaled dataset is the first dataset (version 1.0) developed with such a high spatiotemporal resolution over such a long time period for China.

Compared to the raw data, the downscaled data exhibit smaller deviations and higher spatial resolutions that suggest that the Delta downscaling framework can be used to downscale and correct low-resolution time series climate data. A reference climatology with higher spatial resolution could produce more accurate downscaled data with a higher spatial resolution (Tables 1 and 2). These inferences suggest that (1) the reference climatology is the key factor influencing the accuracy of downscaled data, and thus (2) more accurate reference climatologies should be considered and (3) the accuracy does not depend on the spatial resolution. The Delta downscaling framework involves a basic assumption that a short-term reference climatology can represent the long-term climatology over a region (Brekke et al., 2013; Mosier et al., 2014). In this study, we adopted a reference climatology for the period of 1970–2000 to conduct the downscaling for the period 1901–2017. The evaluations conducted for the period of 1951–2016 (Table 1) demonstrated that the reference climatology adopted was reasonable. The time-series downscaled data with a spatial resolution of 0.5' were almost coincident with the observed data for the periods of 1951–1969 and 2001–2016 (Figs. 6-9). Therefore, the reference climatology should be selected with care and improved if higher accuracy is needed.

The 0.5' downscaled TMP and PRE dataset captures the detailed climatology of the whole of China very well (Fig. 3). It accurately represents climate characteristics such as the minimum TMP at high elevations (e.g., the Qinghai–Tibet Plateau), the maximum TMP at low elevations (e.g., Turpan Basin), and heavy PRE in marine areas (e.g., Taiwan Island). The biases of the climatology were only -0.13 °C for the monthly minimum TMP, -0.53 °C for the monthly maximum TMP, -0.04 °C for the annual mean TMP, and -5.8 mm for the annual total PRE (Table 2). Furthermore, the climatology of 0.5' downscaled data is almost coincident with the observed data for each climatic variable at each weather station (Fig. 4). Therefore, the 0.5' downscaled TMP and PRE dataset is considered to be appropriate for use in assessing climate change and its spatial effects. The 0.5' downscaled TMPs and PRE dataset also represents detailed annual trends in climatic variables over the whole of China very well (Fig. 5). The dataset precisely represents the trends and their significance levels over the geographic space, such as significant increasing and decreasing trends for the maximum TMP and PRE. In general, this dataset captures the annual and seasonal trends very well, except for the autumn trend in the minimum TMP and the summer trend in the maximum TMP (Table 3). Considering that these exceptions have relatively small bias (0.06 °C 10 yr$^{-1}$ and 0.058 °C 10 yr$^{-1}$, respectively) relative to the observed trends, the 0.5' downscaled TMPs and PRE dataset can be used successfully to assess temporal climatic variations and their effects.

As mentioned previously, the accuracy of the reference climatology largely determines the quality of the dataset. In this study, the reference climatology from WorldClim was adopted. Although the evaluation indicted that the quality of the dataset is very good, there is gap between the dataset and the observed data. We think that a new and better reference climatology should be generated using observed data gathered across China. However, the current release of public climate data over China is insufficient to construct a reference climatology better than that available from WorldClim. In ongoing research, we are devoting efforts to collecting more public and private climate data so that we can construct a better





reference climatology and then generate a more accurate downscaled dataset for China. Another limitation is the difficulty of validation. As shown in Fig. 1, most of the weather stations in China are located in the southeastern part of the country, which experiences heavy precipitation and relatively high temperatures; there are fewer stations in high-altitude areas, which have relatively low temperatures, and the northwestern part of China, which receives less precipitation. Thus, it is difficult to

5 evaluate the credibility of the dataset in the high mountains and desert areas. We expect other researchers to validate our product in these areas using different data resources. More validation and applications are welcome. In addition, because of the limitations associated with computational resources, the spatial resolution of the reference climatology, and the raw CRU data, the resolution of this dataset is limited to monthly and a 0.5' (approximately 1 km) grid spacing. However, the current dataset (approximately 378 GB) is huge to process and store. The computational resources and disk usage required for the

10 dataset will increase exponentially as the spatiotemporal resolution increases (Gao et al., 2018). For such a huge amount of data, storage and extraction are not convenient. Supercomputers and parallel computing will be necessary to work with larger such datasets in the future. Another limitation is that the current dataset only includes historical climate data. Many GCM products have been released, but their coarse spatial resolution and low accuracy prevent detailed projections of future climate trends and their effects on local scales, which are pressing needs for planning local strategies to cope with the

15 negative effects of future climate changes. The Delta spatial downscaling procedure has been employed to generate future climate data at high resolutions for some areas (Peng et al., 2017). The issues associated with computational resources, validation, and a reasonable reference climatology must be addressed to generate high-resolution climate data for China in the future. Higher-resolution data, more validation, and a better reference climatology for historical and future climate data (version 2.0) will all be concerns in our future research.

**Supplement**

**Table S1:** Statistical characteristics of observed and raw or downscaled monthly TMPs and PRE for 1950–2016 at 745 weather stations. The values shown are standard deviations of the evaluation results for the weather stations.

**Competing interests**

The authors declare that they have no conflict of interest.

**Acknowledgements**

This study was supported jointly by the National Natural Science Foundation of China (41601058 & U1703124) and the CAS Light of West China Program (XAB2015B07).



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



**Table 1:** Statistical characteristics of observed and raw or downscaled monthly TMP and PRE values for 1951 to 2016 for the 745 weather stations. The values shown are the average results for the weather stations. Their standard deviations are listed in Table S1.

| | Res | $MAE_c$ | $MAE_l$ | $MAE_n$ | $RMSE_c$ | $RMSE_l$ | $RMSE_n$ | $NSE_c$ | $NSE_l$ | $NSE_n$ |
|---|---|---|---|---|---|---|---|---|---|---|
| Minimum | 30' | 1.624 | | | 1.893 | | | 0.891 | | |
| TMP (°C) | 10' | 1.603 | 1.601 | 1.602 | 1.824 | 1.823 | 1.825 | 0.893 | 0.894 | 0.893 |
| | 5' | 1.342 | 1.341 | 1.342 | 1.565 | 1.564 | 1.567 | 0.928 | 0.931 | 0.928 |
| | 2.5' | 1.198 | 1.196 | 1.197 | 1.423 | 1.421 | 1.423 | 0.951 | 0.954 | 0.951 |
| | 0.5' | 1.145 | 1.143 | 1.145 | 1.373 | 1.372 | 1.375 | 0.959 | 0.961 | 0.959 |
| Mean | 30' | 1.384 | | | 1.539 | | | 0.871 | | |
| TMP (°C) | 10' | 1.226 | 1.225 | 1.227 | 1.386 | 1.384 | 1.387 | 0.898 | 0.899 | 0.898 |
| | 5' | 1.016 | 1.013 | 1.015 | 1.178 | 1.175 | 1.180 | 0.934 | 0.936 | 0.935 |
| | 2.5' | 0.873 | 0.870 | 0.874 | 1.038 | 1.033 | 1.040 | 0.963 | 0.967 | 0.964 |
| | 0.5' | 0.844 | 0.842 | 0.843 | 1.009 | 1.006 | 1.011 | 0.972 | 0.974 | 0.972 |
| Maximum | 30' | 1.882 | | | 2.355 | | | 0.737 | | |
| TMP (°C) | 10' | 1.845 | 1.843 | 1.846 | 2.067 | 2.065 | 2.068 | 0.748 | 0.749 | 0.747 |
| | 5' | 1.536 | 1.535 | 1.538 | 1.760 | 1.758 | 1.763 | 0.829 | 0.830 | 0.829 |
| | 2.5' | 1.327 | 1.326 | 1.330 | 1.551 | 1.550 | 1.553 | 0.898 | 0.900 | 0.899 |
| | 0.5' | 1.226 | 1.224 | 1.228 | 1.449 | 1.447 | 1.452 | 0.921 | 0.923 | 0.922 |
| PRE | 30' | 15.128 | | | 23.567 | | | 0.627 | | |
| (mm) | 10' | 13.736 | 13.636 | 13.746 | 21.063 | 20.926 | 21.071 | 0.737 | 0.739 | 0.737 |
| | 5' | 13.735 | 13.635 | 13.775 | 21.044 | 20.908 | 21.108 | 0.747 | 0.749 | 0.746 |
| | 2.5' | 13.730 | 13.630 | 13.746 | 21.007 | 20.872 | 21.089 | 0.757 | 0.758 | 0.755 |
| | 0.5' | 13.697 | 13.596 | 13.743 | 20.972 | 20.836 | 21.045 | 0.758 | 0.759 | 0.757 |

Notes: Res indicates the spatial resolution. The subscripts $c$, $l$, and $n$ indicate bicubic, bilinear, and nearest-neighbor interpolations, respectively. The raw TMPs and PRE are at a spatial resolution of 30' and were compared directly to the observed data.





**Table 2:** Comparison of the averaged climatology among the 745 weather stations, based on observed and downscaled values for four spatial resolutions, along with raw values, for 1951 to 2016.

| Resolution | Monthly minimum TMP (°C) | Monthly maximum TMP (°C) | Annual mean TMP (°C) | Annual total PRE (mm) |
|---|---|---|---|---|
| 30' | -9.88 ± 11.49 | 29.43 ± 4.97 | 11.21 ± 6.82 | 849.9 ± 491.1 |
| 10' | -9.82 ± 11.24 | 30.10 ± 4.71 | 11.33 ± 6.67 | 861.9 ± 483.6 |
| 5' | -9.50 ± 11.26 | 30.38 ± 4.47 | 11.65 ± 6.56 | 866.2 ± 485.1 |
| 2.5' | -9.31 ± 11.27 | 30.59 ± 4.20 | 11.93 ± 6.45 | 868.3 ± 487.0 |
| 0.5' | -9.29 ± 11.20 | 30.66 ± 4.18 | 11.78 ± 6.39 | 872.7 ± 486.5 |
| Observation | -9.16 ± 11.50 | 31.19 ± 4.40 | 11.82 ± 6.36 | 878.5 ± 517.5 |

Notes: Monthly minimum and maximum TMPs are 1% and 99% quantile values, respectively, based on monthly time-series data. Annual total PRE and mean TMP values were calculated for full years.



**Table 3:** Trends in annual and seasonal TMP and PRE values for the 745 weather stations from 1951 to 2016.

| | | Annual | Spring | Summer | Autumn | Winter |
|---|---|---|---|---|---|---|
| Minimum TMP (°C 10 yr$^{-1}$) | Observation | 0.323** | 0.255** | 0.157** | 0.162* | 0.324** |
| | Downscaled | 0.277** | 0.235** | 0.089** | 0.102 | 0.277** |
| | Raw | 0.263** | 0.217** | 0.092** | 0.096 | 0.264** |
| Maximum TMP (°C 10 yr$^{-1}$) | Observation | 0.062 | 0.097* | 0.021 | 0.073 | 0.140 |
| | Downscaled | 0.015 | 0.033* | -0.037 | 0.016 | 0.077 |
| | Raw | 0.001 | 0.001 | -0.046 | 0.001 | 0.070 |
| Mean TMP (°C 10 yr$^{-1}$) | Observation | 0.167** | 0.196** | 0.077** | 0.154** | 0.211** |
| | Downscaled | 0.108** | 0.133** | 0.030* | 0.088* | 0.158** |
| | Raw | 0.101** | 0.119** | 0.001 | 0.073 | 0.149** |
| PRE (mm 10 yr$^{-1}$) | Observation | 2.498 | -2.079 | 0.488 | -0.766 | 1.655 |
| | Downscaled | 2.985 | -2.673 | 0.781 | -1.047 | 1.163 |
| | Raw | 3.341 | -2.849 | 1.389 | -1.112 | 1.027 |

Notes: ** indicates 99 % significance level; * indicates 95 % significance level.

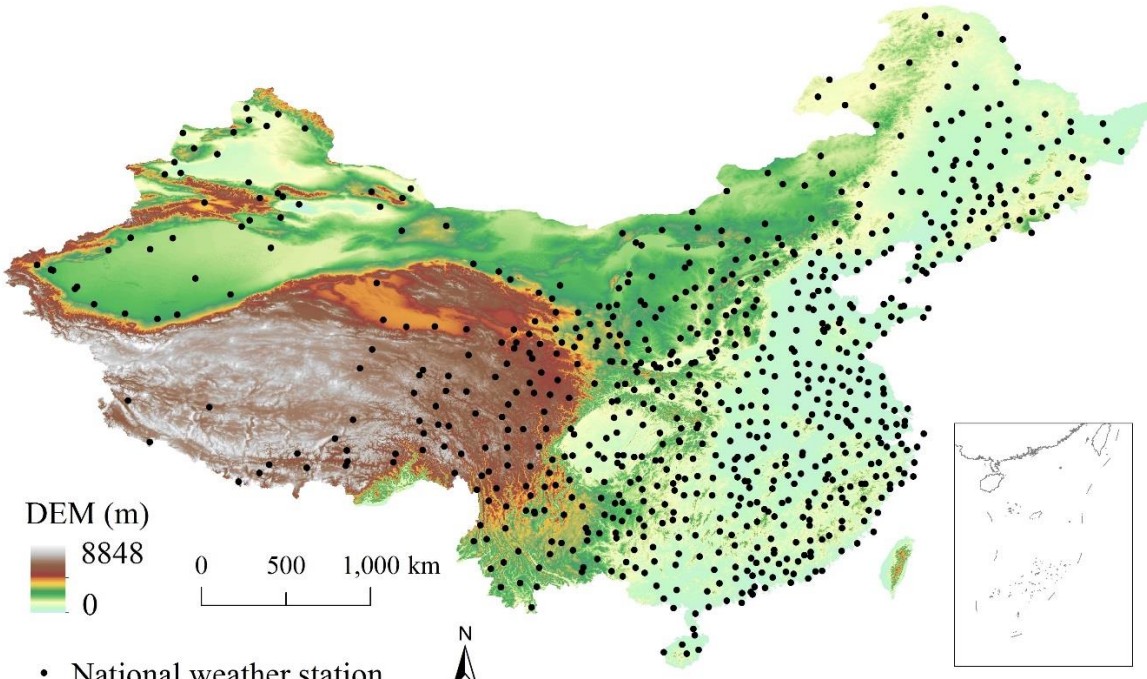

**Figure 1:** Spatial distribution of the national weather stations across China.

PRE in July 2017 with 30' from CRU

Mean PRE in July from 1970 to 2000
with 30' from CRU
(a)

Anomaly with 30'
(b)

Anomaly with 0.5'
(c)

Interpolated

Mean PRE in July from 1970 to 2000
with 0.5' from WorldClim

Downscaled PRE in July 2017 with 0.5'
(d)

**Figure 2:** Schematic illustration of the Delta spatial downscaling process using the PRE in July 2017 obtained from the CRU data as an example.

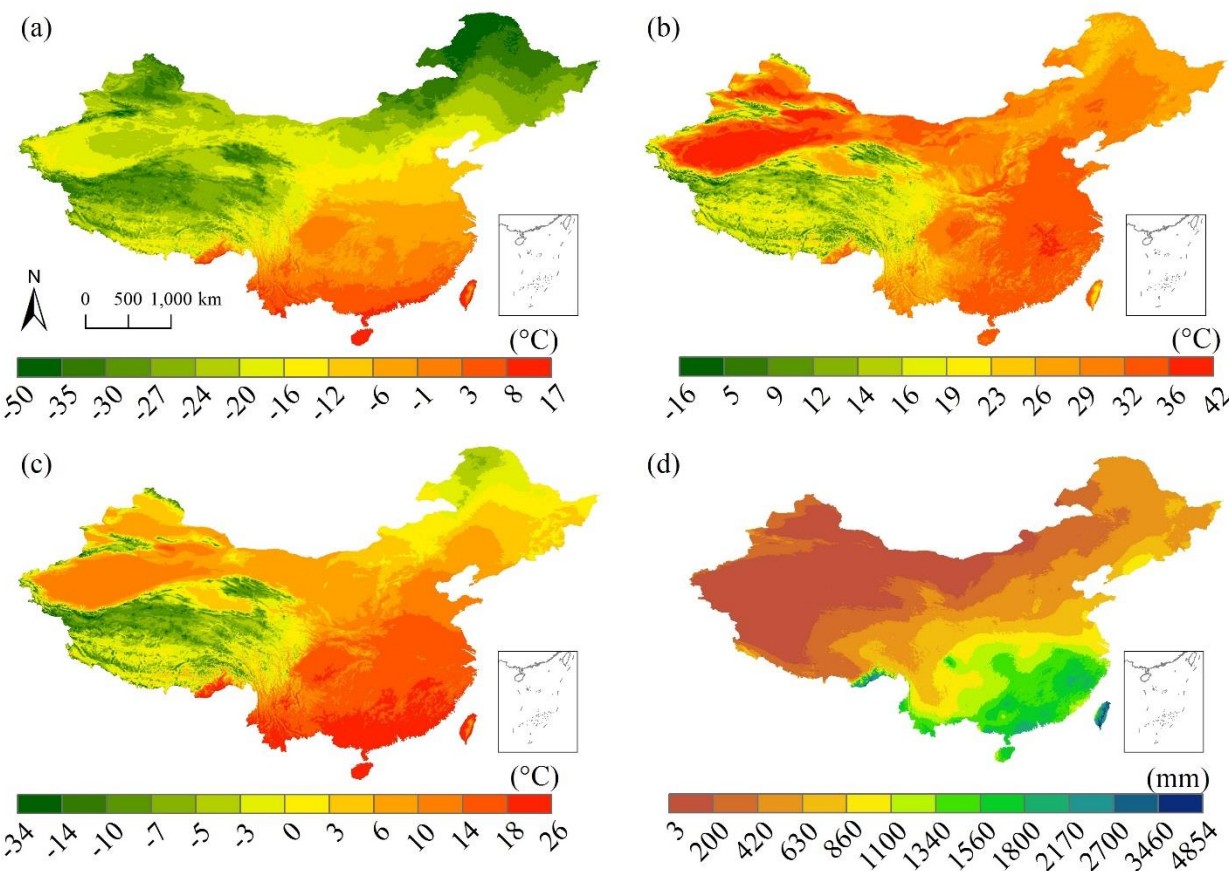

**Figure 3:** Spatial distributions of the climatology in TMP and PRE values over China at a spatial resolution of 0.5' for 1901 to 2017. (**a**) and (**b**) are the 1 % and 99 % quantiles of monthly minimum and maximum temperatures from 1901 to 2017, respectively; (**c**) and (**d**) are the average annual mean temperature and total precipitation, respectively, from 1901 to 2017.



**Figure 4:** Scatterplots of observed (*x*) and downscaled/raw (*y*) climatology values for 1951 to 2016 at the 745 weather stations. Minimum and maximum TMPs are the 1 % and 99 % quantiles of monthly minimum and maximum temperatures, respectively, during 1951–2016; mean TMP and PRE are the average annual mean temperature and total precipitation, respectively, during 1951–2016.

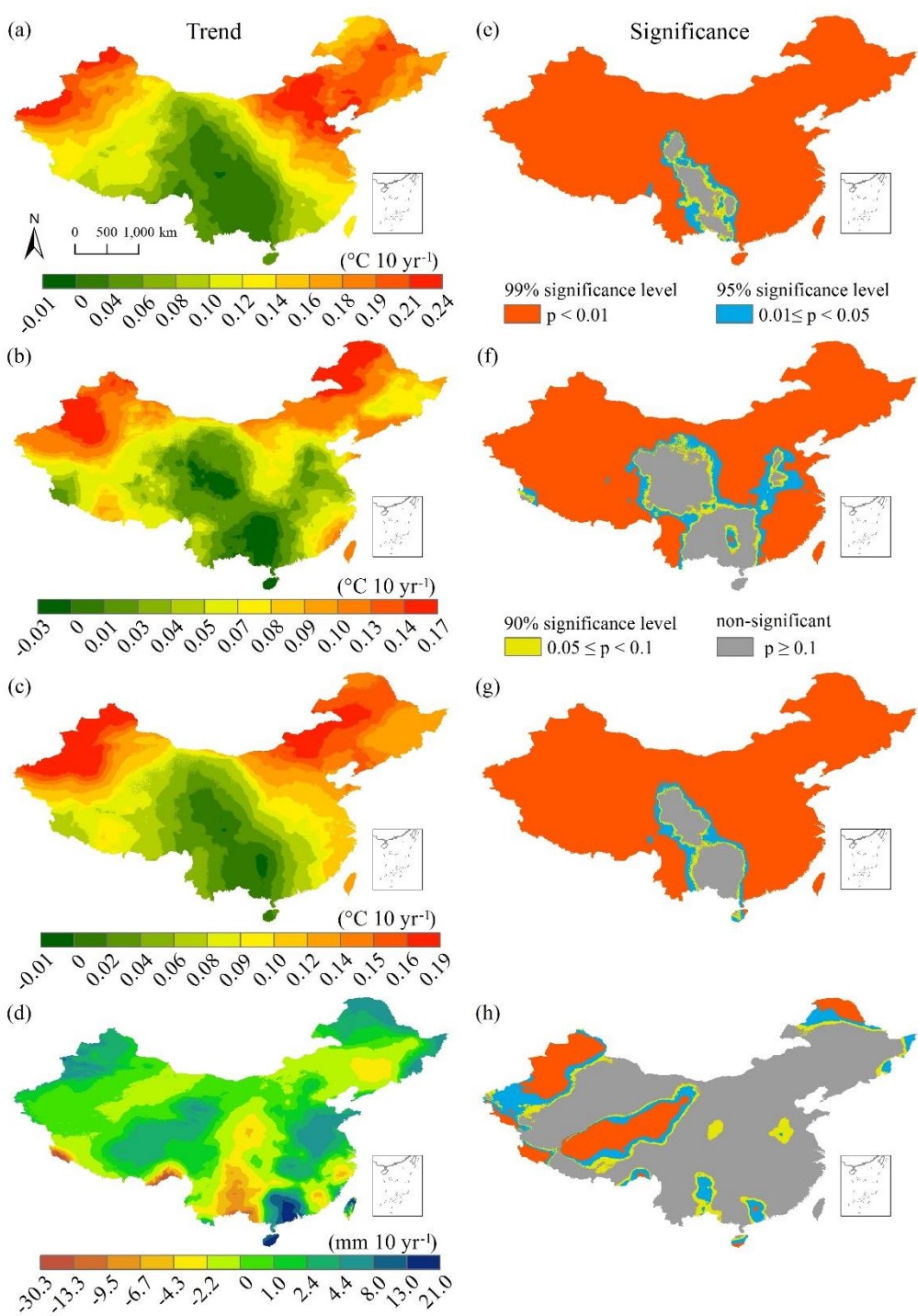

**Figure 5:** Spatial distributions of the annual trends in TMP and PRE values and their significance levels across China at a spatial resolution of 0.5' for 1901 to 2017. (**ae**), (**bf**), (**cg**), and (**dh**) are the annual minimum TMP, maximum TMP, mean TMP, and PRE, respectively. Annual minimum and maximum TMPs are the 1 % and 99 % quantiles, respectively.

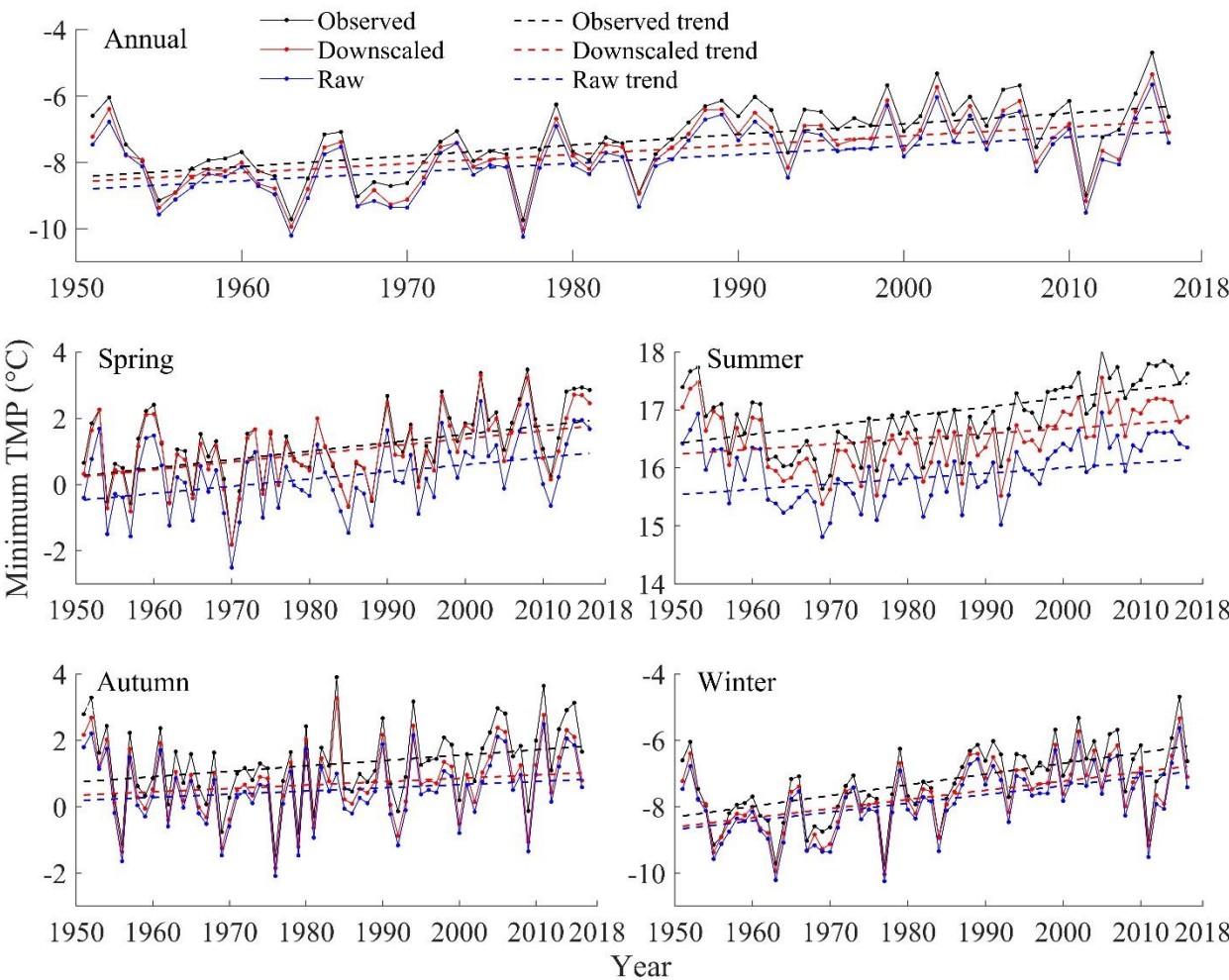

**Figure 6:** Temporal variations in annual and seasonal minimum TMP values for 745 weather stations during 1951–2016.

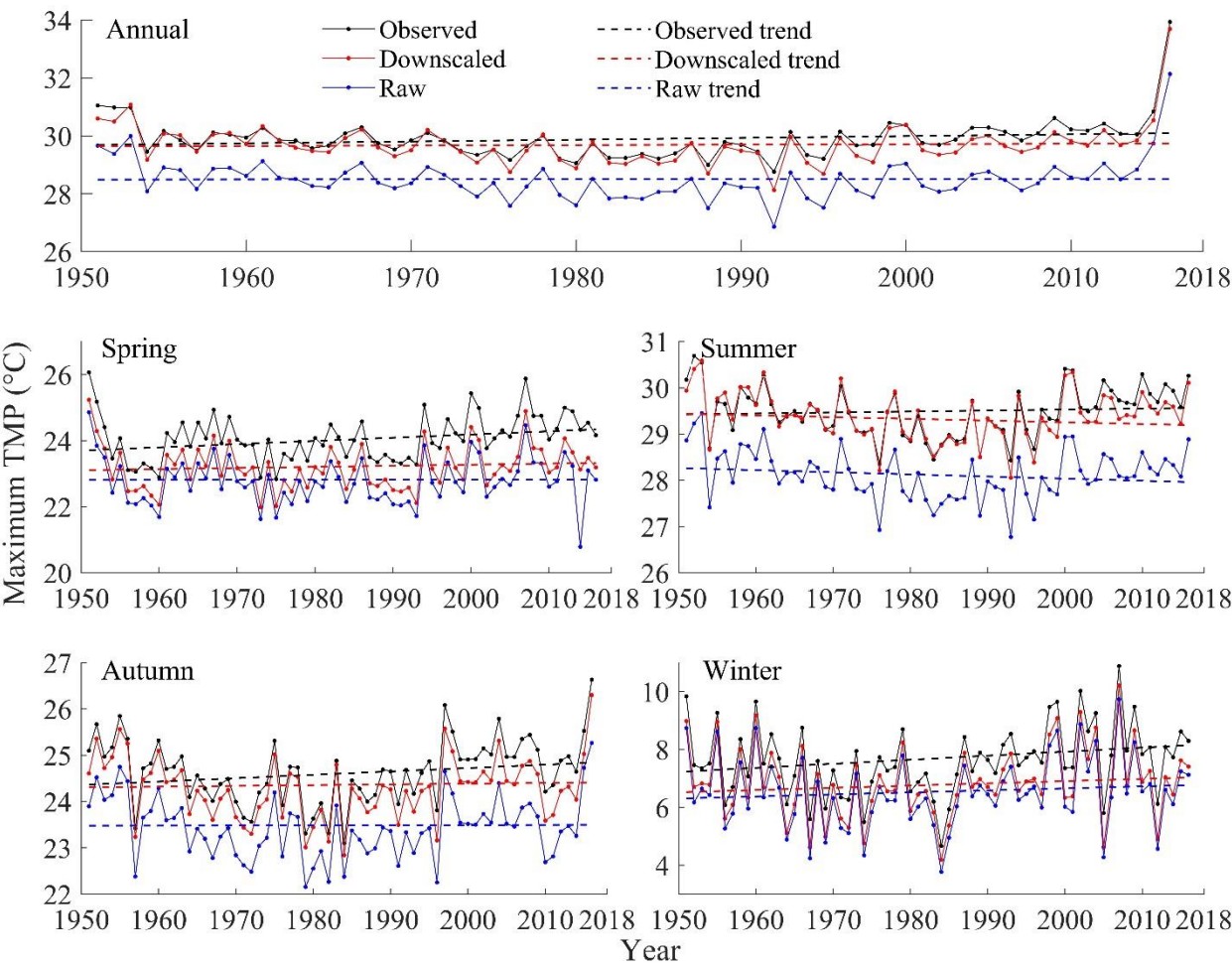

**Figure 7:** Temporal variations in annual and seasonal maximum TMP values for 745 weather stations during 1951–2016.

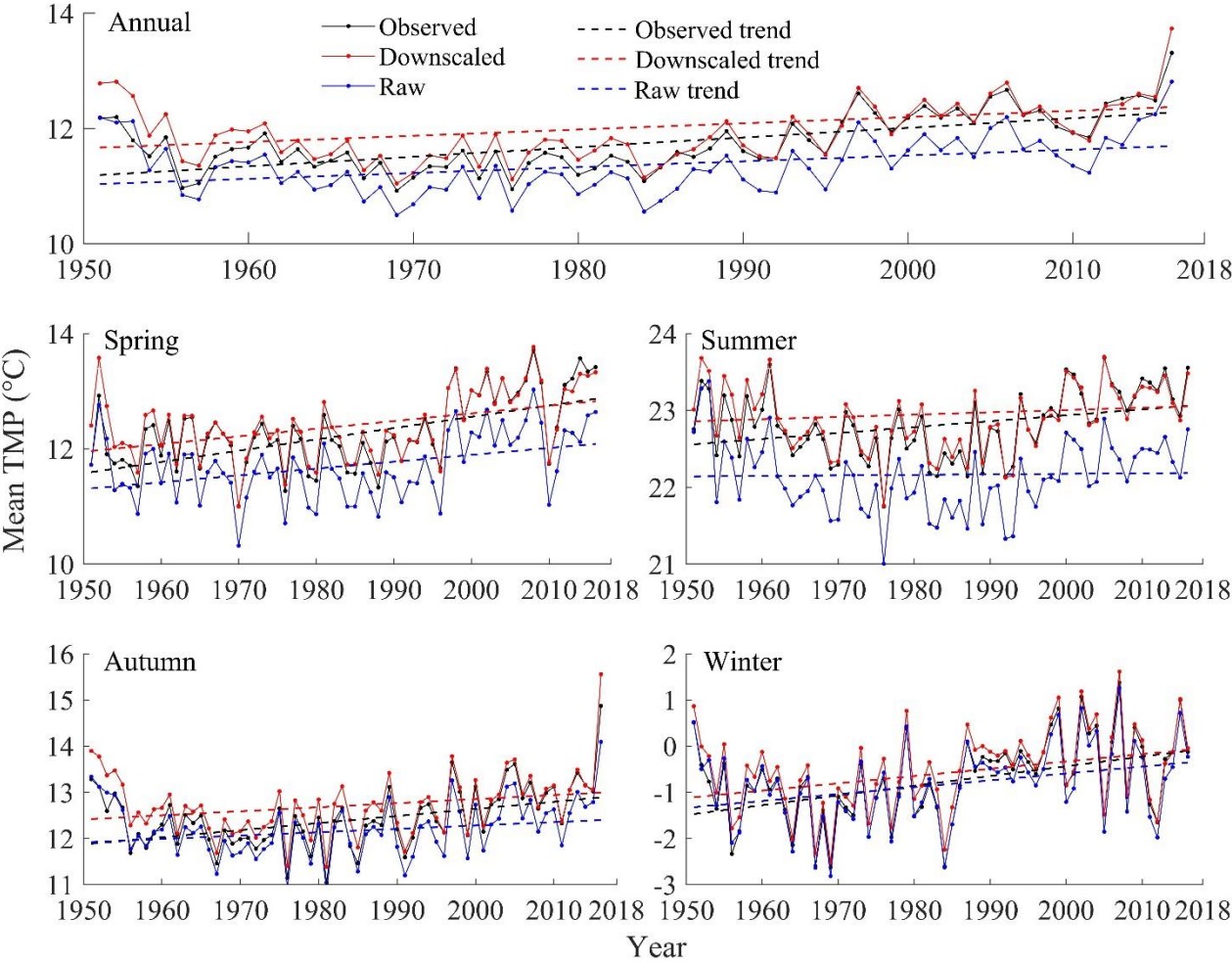

**Figure 8:** Temporal variations in annual and seasonal mean TMP values for 745 weather stations during 1951–2016.

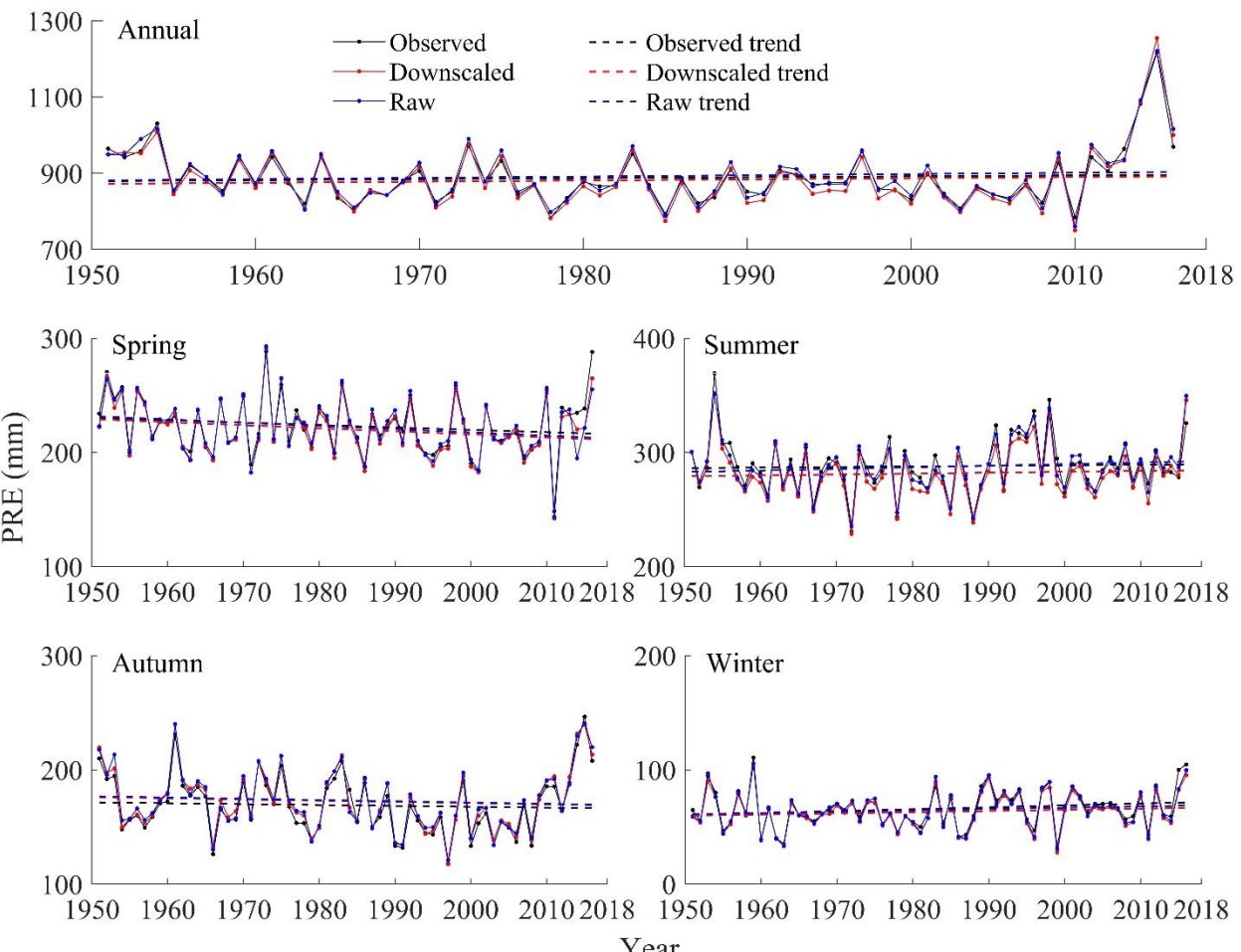

**Figure 9:** Temporal variations in annual and seasonal PRE values for 745 weather stations during 1951−2016.