# Peer review of "High-spatial-resolution monthly temperature and precipitation dataset for China for 1901–2017"

_Earth System Science Data, 2019_

## Referee Comment (RC1) · Anonymous Referee #1 · 3 Jul 2019

The authors downscaled the monthly CRU temperature and precipitation data in 30' grids into 0.5' and compared against 745 weather station observations over China. They concluded that the down scaled dataset is closer to the observations than the original CRU dataset. The analysis and presentation are very clear, but their motivation of the analysis and the liability of the downscaled data are questionable. Their study is more like an analysis and comparison of the CRU dataset rather than an original creation of a new data set. Therefore I am not in favor of publishing their analysis in the ESSD data journal.

Major comments:

(1) It is not clear what temperature it is in their analysis. Is it land surface air temperature at 2m or surface temperature over land (0m)? Given current observation capability,

how is it possible to generate 1km TMP and PRE datasets?

(2) How is the WorldClim data in 0.5' created and what observations are used? What is its reliability, how many station data is used over China (at most 745 statios)? As shown in Figure 2 (assuming the same color scale is used), the climatology of PRE (and TMP as shown later in Figures 6‒8) in CRU is very different from WorldClim. I am wondering CRU data may have systematic climatological drift, the large difference between CRU and observations may mostly arise from its climatological drift. If this is the case, the downscaling may not help reduce the error as authors concluded. My suggestion is to include an additional analysis of using 30' WorldClim without downscaling and compare it with that downscaled at 0.5'. If this not the case, the downscaled data is indeed better than the CRU data, authors should address the reasons why it is better.

(3) As shown in Table 2, the uncertainty (±values) are very large, which is much larger than the differences between observed and downscaled mean values. Therefore, it is very likely that the difference between observations and downscaled data is statistically insignificant unless the authors can prove that is indeed the case.

(4) As shown in Table 3, the authors focused on the statistical significance of the trends, but they ignored the more important question whether the differences among CRU, downscaled data, and observations are statistically significant. I suggest the authors using uncertainty (±values) instead of "**" marking.

(5) Figure 4 and its discussion in the main text: Left and Right columns should be explained in the figure caption. I am wondering whether the correlations are mostly associated with climatologies. It should be more convincing if anomalies are used in the diagrams.

Minor comments

P3L25, Delta downscaling, a reference is needed and a brief description is helpful.

P3L28‒30, The calculation of TMP anomaly is conventional, but why PRE anomaly

is defined by ratio? What happens if the difference is defined for PRE?

P4L10, NSE needs a reference.

P4L16, "raw" data, CRU data can never be called "raw". How many station data are used in CRU over China? If all 745 station data are used in CRU, the comparison in Section 4.1 is not independent!

P5, Section 4.2, first paragraph, it is not clear whether the description is for the downscaled data of 0.5'. I also suggest use the same color scale for the TMPs in Figure 3. Second paragraph, see the major comments (2).

P6L5, "downward" trend, check and verify it.

Figure 5 (and Figure 3), the focus should be the difference between CRU and downscaled data rather than the trend (and climatology) itself.

---

## Referee Comment (RC2) · Anonymous Referee #2 · 15 Jul 2019

This study by Peng et al. developed a high-resolution and long-term climate dataset over China. The CRU data was downscaled to 1km using the Delta downscaling framework. The topic is interesting, and the product would be useful in climate-related studies for the nation. However, I think the paper needs some improvement and further discussion before it can be published.

My major concerns include:

1. The Delta downscaling improves the spatial representation of temperature/precipitation climatology using high-resolution WorldClim as the reference climatology. However, it is hard to understand how this downscaling method improves temporal variability (or trend), because the temporal change is simply based on the interpolated anomalies from low-resolution CRU. This limitation should be at least explained

and discussed in the manuscript.

2. The downscaled dataset is developed based on the WorldClim reference climatology from 1970 to 2000, and the data evaluation is performed for the period 1951-2016. Can the authors first evaluate the reference data (WorldClim) at different resolutions? Also, because there is an overlapping period for data training and evaluation, is it possible to use two separate periods, in which one is for downscaling and the other one is for data evaluation?

3. The authors need to discuss the possible reasons why CRU temperatures have systematic cold biases.

4. The dataset covers from 1901 to 2017, but most of the evaluations and discussion are about post-1950. Data quality or uncertainties before 1950 need more discussion.

Specific Comments:

1. P3, L10-L15: Can we get the information about how many stations in China were used for CRU TS and WorldClim? How different are they? Are they comparable to the 745 weather stations used in this study?

2. P4, L17-19: I assume the final product is generated using the bilinear interpolation method? This should be mentioned in conclusion and abstract as well.

3. P5, section 4.3: Trend is one aspect of the temporal variations. It would be better to also calculate the correlation of the time series.

4. Table 1: These metrics are applied to the climatology of TMP and PRE for 1951-2016? Or applied to time series of monthly TMP and PRE, then averaged over the 745 stations? Or any other way? This should be clarified in the main text or table caption. Same issue for Table S1.

5. Figure 4: Becuase the climatology is “corrected” using high-resolution reference data, it is not surprising that the downscaled data outperformed the CRU data in terms

of the climatology. As suggested in Comment #3, it would be better to have a similar figure to show the time series (or anomalies).

6. Figures 6-9: These figures are not quite informative. It would be better to add the trends as text on the figures. For figure 9, it is really hard to distinguish those three lines.

---

## Referee Comment (RC3) · Anonymous Referee #3 · 20 Jul 2019

The authors proposed a high-spatial-resolution monthly temperature and precipitation dataset for China by Delta downscaling of CRU dataset. The original CRU at 30' resolution is downscaled to 1km grid. The new downscaled data set include four common climate elements that are always the driven data for various models. This topic is quite interesting and would be useful for the climate change community. However, there are some obvious flaws in the downscaling procedure and the evaluation part. More interpretation and discussion should be improved. Therefore, I do not support this publication in ESSD at current version.

General comments:

1. Downscaling is a complicated procedure, especially for precipitation from 30' to 1km grid. I do not agree that the downscaled data set represents the local physical

[Figure]

**[ESSDD**
process. Actually, Delta downscaling is an interpolation method. CRU data set is also actually produced by interpolation method. The final downscaling result is the sum of "raw" CRU and interpolated anomaly. For my understanding, there is not any physical process involved. Conventionally, for a better local representation, local topography features should be considered such as aspect, slope and elevation.

2. WorldClim data set is used as the reference data in downscaling. However, how well does WorldClim represent the climatology over China? I did not find this information in the current version. The bias of WorldClim could be transferred into the final results. Therefore, it is not easy to understand why the downscaled data has a better performance. If the authors use other reference data, how will the downscaling result be?

3. In addition, the "Direct evaluation" is not adequate. The time series are different for CRU, WorldClim, and observation. How do the authors guarantee the consistency of time series, in particular the period 1901-1950? Meanwhile, the mean climatology is calculated from 1970-2000. Is this time period appropriate for representation? For precipitation, the observation has shown significant nonstationary features after 1980s in China under the global warming. Unfortunately, Delta downscaling method does not consider the nonstationary.

4. The authors evaluated the new data set using 745 observations over China. I think it is not enough, especially for the west of China, such as the high mountains areas and Tibet Plateau. Meanwhile, most observations begin after 1950, how about the pre-1950? Therefore, it is hard to conclude the data set is "sufficiently reliable".

5. How many observations have been used in CRU and WorldClim? These sites should be excluded since they destroy the independence of evaluation.

Specific comments:

1. Figure 1, the range of DEM from 0 to 8848 is wrong. The Turpan Basin is for sure

below the mean sea level. What is the spatial resolution of DEM in this figure?

2. Figure2, it is hard to follow this downscaling framework. There is no legend for all figures, which is the mandatory element. The color scales should be the same for a better comparison.

3. More interpretation should be given for the Delta downscaling method. For example, how to calculate the "ratio" for PRE anomaly? Is there a simple mathematical formula?

4. Once again, "Direct evaluation" is not sufficient. More details about the bias or errors should be supplemented.

---

## Author Comment (AC1) · 3 Aug 2019

The authors downscaled the monthly CRU temperature and precipitation data in 30' grids into 0.5' and compared against 745 weather station observations over China. They concluded that the down scaled dataset is closer to the observations than the original CRU dataset. The analysis and presentation are very clear, but their motivation of the analysis and the liability of the downscaled data are questionable. Their study is more like an analysis and comparison of the CRU dataset rather than an original creation of a new data set. Therefore I am not in favor of publishing their analysis in the ESSD data journal.

Response: We did develop a new dataset, and the developed dataset is novel for its high spatial resolution and long period. In specific, based on a spatial downscaling technique and three data sources (i.e. CRU grid data, WorldClim grid data and site-specific data from weather stations), we created a dataset of 0.5' for the period 1901-2017. As far as we know, this is the dataset with the highest spatial resolution and the longest period over China. The new dataset greatly improved the accuracy of the original datasets, especially for those regions with limited observations. The dataset is thus prominent as inputs for ecological models such as dynamic vegetation models.

Major comments:

(1) It is not clear what temperature it is in their analysis. Is it land surface air temperature at 2m or surface temperature over land (0m)? Given current observation capability, how is it possible to generate 1km TMP and PRE datasets?

Response: Yes, you are right. Consistent with CRU and WorldClim, this study employed the land surface air temperature at 2 m. As you mentioned, it is hard to generate 1-km climate datasets with observation; however, the 1-km datasets are highly desirable for understanding climate-related natural processes. This is actually the motivation of this study, which is also the novelty of this study.

We spatially downscaled the latest datasets from CRU and WorldClim to generate a long-term dataset of 1-km spatial resolution. We did not incorporate our own site observational data, but carried out direct interpolation of low-resolution data considering the orographic effects, the distance to coast and satellite-derived covariate effects. The method we used for data generation is similar as the other data centers. Further, our validation based on site observation independent of the original datasets can show the reliability of our datasets. The results in the manuscript confirmed the applicability of our method.

(2) How is the WorldClim data in 0.5' created and what observations are used? What is its reliability, how many station data is used over China (at most 745 statios)? As shown in Figure 2 (assuming the same color scale is used), the climatology of PRE (and TMP as shown later in Figures 6  Š8) in CRU is very different from WorldClim. I am wondering CRU data may have systematic climatological drift, the large difference between CRU and observations may mostly arise from its climatological drift. If this is the case, the downscaling may not help reduce the error as authors concluded. My suggestion is to include an additional analysis of using 30' WorldClim without downscaling and compare it with that downscaled at 0.5'. If this not the case, the downscaled data is indeed better than the CRU data, authors should address the reasons why it is better.

Response: Thanks for this suggestion. Our new version will address your concerns as follows.

The WorldClim datasets used in this study have four spatial resolutions (10', 5', 2.5', and 0.5'). They were created using 9000–60000 weather stations over the globe based on the thin-plate splines interpolation method. The interpolation considered covariation with the latitude, longitude, elevation, distance to the nearest coast, and three satellite-derived covariates: the maximum and minimum land surface temperature and cloud cover, obtained from the MODIS satellite platform. The cross-validation correlations indicated that the WorldClim datasets held good performance around the world, because of the introduction of satellite-derived and distance to the nearest coast covariates (Fick and Hijmans, 2017). The WorldClim datasets used data of 323 weather stations across China (Fick and Hijmans, 2017) (Figure 1).

As for the possible systematic climatological drift, we first evaluated the reliability of the WorldClim datasets (Tables 1 and 2) with 496 weather stations independent of the 323 stations for the generation of WorldClim data (Figure 1). Overall, the WorldClim datasets have high performance to represent the monthly climatology over China region, and the dataset performs better for higher spatial resolution. In specific, the absolute errors become smaller with increasing spatial resolution (Table 1) and the correlations get greater with increasing spatial resolution (Table 2).

[Figure]

- ▲ Stations for producing the CRU TS and WorldClim data (323 )
- • Stations for validating the downscaled results in this study (496)

Figure 1. Spatial distribution of the national weather stations across China.

Table 1. The mean absolute errors between the observed and WorldClim climatology at different spatial resolutions over the 496 weather stations. The period ranges from 1970 to 2000.

|  |  | Jan | Feb | Mar | Apr | May | Jun | Jul | Aug | Sep | Oct | Nov | Dec |
|---|---|---|---|---|---|---|---|---|---|---|---|---|---|
| Minimum | 10' | 0.726 | 0.675 | 0.615 | 0.533 | 0.515 | 0.533 | 0.789 | 0.759 | 0.719 | 0.639 | 0.643 | 0.656 |
| TMP (°C) | 5' | 0.653 | 0.596 | 0.521 | 0.467 | 0.450 | 0.429 | 0.660 | 0.633 | 0.607 | 0.523 | 0.514 | 0.550 |
|  | 2.5' | 0.632 | 0.563 | 0.484 | 0.433 | 0.411 | 0.372 | 0.602 | 0.574 | 0.543 | 0.459 | 0.449 | 0.503 |

|  |  | Jan | Feb | Mar | Apr | May | Jun | Jul | Aug | Sep | Oct | Nov | Dec |
|---|---|---|---|---|---|---|---|---|---|---|---|---|---|
|  | 0.5' | 0.622 | 0.549 | 0.474 | 0.430 | 0.408 | 0.354 | 0.567 | 0.541 | 0.513 | 0.428 | 0.420 | 0.484 |
| Mean TMP (°C) | 10' | 0.450 | 0.481 | 0.470 | 0.482 | 0.487 | 0.478 | 0.455 | 0.445 | 0.427 | 0.425 | 0.425 | 0.427 |
|  | 5' | 0.401 | 0.426 | 0.385 | 0.390 | 0.400 | 0.391 | 0.379 | 0.387 | 0.380 | 0.367 | 0.362 | 0.377 |
|  | 2.5' | 0.365 | 0.378 | 0.338 | 0.332 | 0.351 | 0.342 | 0.338 | 0.356 | 0.348 | 0.333 | 0.331 | 0.349 |
|  | 0.5' | 0.355 | 0.366 | 0.328 | 0.322 | 0.337 | 0.330 | 0.334 | 0.351 | 0.343 | 0.331 | 0.324 | 0.342 |
| Maximum TMP (°C) | 10' | 0.832 | 0.821 | 0.809 | 0.909 | 0.827 | 0.678 | 0.718 | 0.734 | 0.644 | 0.658 | 0.630 | 0.687 |
|  | 5' | 0.727 | 0.711 | 0.666 | 0.760 | 0.687 | 0.560 | 0.645 | 0.658 | 0.568 | 0.561 | 0.511 | 0.576 |
|  | 2.5' | 0.664 | 0.637 | 0.591 | 0.670 | 0.597 | 0.485 | 0.589 | 0.600 | 0.531 | 0.509 | 0.447 | 0.517 |
|  | 0.5' | 0.631 | 0.596 | 0.544 | 0.611 | 0.544 | 0.445 | 0.574 | 0.578 | 0.516 | 0.484 | 0.405 | 0.479 |
| PRE (mm) | 10' | 2.165 | 1.869 | 3.476 | 4.662 | 5.651 | 8.416 | 9.716 | 7.993 | 5.825 | 3.968 | 2.202 | 1.378 |
|  | 5' | 2.077 | 1.834 | 3.407 | 4.641 | 5.637 | 8.291 | 9.702 | 7.841 | 5.805 | 3.908 | 2.183 | 1.348 |
|  | 2.5' | 2.074 | 1.813 | 3.404 | 4.603 | 5.594 | 8.268 | 9.664 | 7.705 | 5.742 | 3.904 | 2.182 | 1.334 |
|  | 0.5' | 2.072 | 1.797 | 3.360 | 4.495 | 5.564 | 8.190 | 9.630 | 7.651 | 5.699 | 3.895 | 2.170 | 1.300 |

Table 2. The correlation coefficients between the observed and WorldClim climatology at different spatial resolutions over the 496 weather stations. The period ranges from 1970 to 2000.

|  |  | Jan | Feb | Mar | Apr | May | Jun | Jul | Aug | Sep | Oct | Nov | Dec |
|---|---|---|---|---|---|---|---|---|---|---|---|---|---|
| Minimum TMP (°C) | 10' | 0.987 | 0.984 | 0.977 | 0.969 | 0.963 | 0.962 | 0.955 | 0.957 | 0.956 | 0.971 | 0.984 | 0.987 |
|  | 5' | 0.989 | 0.987 | 0.983 | 0.977 | 0.973 | 0.973 | 0.964 | 0.966 | 0.968 | 0.980 | 0.990 | 0.991 |
|  | 2.5' | 0.989 | 0.988 | 0.985 | 0.981 | 0.978 | 0.977 | 0.968 | 0.971 | 0.974 | 0.985 | 0.992 | 0.992 |
|  | 0.5' | 0.989 | 0.989 | 0.986 | 0.983 | 0.981 | 0.980 | 0.972 | 0.974 | 0.977 | 0.988 | 0.993 | 0.993 |
| Mean TMP (°C) | 10' | 0.986 | 0.979 | 0.968 | 0.955 | 0.949 | 0.949 | 0.956 | 0.958 | 0.966 | 0.974 | 0.982 | 0.987 |
|  | 5' | 0.991 | 0.986 | 0.980 | 0.969 | 0.962 | 0.959 | 0.963 | 0.965 | 0.973 | 0.983 | 0.989 | 0.991 |
|  | 2.5' | 0.993 | 0.990 | 0.986 | 0.977 | 0.970 | 0.965 | 0.968 | 0.970 | 0.978 | 0.986 | 0.992 | 0.993 |
|  | 0.5' | 0.994 | 0.992 | 0.989 | 0.981 | 0.973 | 0.968 | 0.970 | 0.972 | 0.980 | 0.988 | 0.993 | 0.995 |
| Maximum TMP (°C) | 10' | 0.958 | 0.946 | 0.920 | 0.892 | 0.889 | 0.899 | 0.893 | 0.890 | 0.935 | 0.957 | 0.968 | 0.974 |
|  | 5' | 0.969 | 0.961 | 0.946 | 0.921 | 0.912 | 0.912 | 0.898 | 0.896 | 0.939 | 0.965 | 0.978 | 0.982 |
|  | 2.5' | 0.976 | 0.971 | 0.960 | 0.941 | 0.930 | 0.925 | 0.910 | 0.909 | 0.945 | 0.971 | 0.984 | 0.986 |
|  | 0.5' | 0.979 | 0.976 | 0.968 | 0.951 | 0.940 | 0.932 | 0.913 | 0.912 | 0.946 | 0.973 | 0.988 | 0.989 |
| PRE (mm) | 10' | 0.976 | 0.980 | 0.978 | 0.979 | 0.974 | 0.961 | 0.903 | 0.920 | 0.941 | 0.908 | 0.939 | 0.965 |
|  | 5' | 0.976 | 0.980 | 0.979 | 0.979 | 0.974 | 0.961 | 0.905 | 0.924 | 0.943 | 0.911 | 0.940 | 0.966 |
|  | 2.5' | 0.976 | 0.981 | 0.980 | 0.979 | 0.974 | 0.962 | 0.908 | 0.930 | 0.943 | 0.913 | 0.941 | 0.967 |
|  | 0.5' | 0.977 | 0.981 | 0.981 | 0.980 | 0.975 | 0.962 | 0.909 | 0.930 | 0.944 | 0.914 | 0.941 | 0.968 |

Further, we evaluated the original CRU data and validated the downscaled datasets of 10', 5', 2.5', and 0.5' with the 496 stations independent of those stations for original dataset generation (Table 3). The results also indicated that downscaled datasets had better performance than the original CRU dataset, especially for the 0.5' dataset.

Considering the above two validations, the employed original data had good performance and our downscaled data even improved the performance since we presented data of even higher resolution. It appeared the systematic climatological drifts do not exist or have little impacts on data quality and our technique further decreased them if they exist.

Table 3. Statistical characteristics between original/downscaled CRU and observed monthly TMPs and PRE in the time series (1951–2016). The values shown here are the averaged evaluation results at all 496 weather stations.

| | Res | $MAE_c$ | $MAE_l$ | $MAE_n$ | $RMSE_c$ | $RMSE_l$ | $RMSE_n$ | $NSE_c$ | $NSE_l$ | $NSE_n$ | $Cor_c$ | $Cor_l$ | $Cor_n$ |
|---|---|---|---|---|---|---|---|---|---|---|---|---|---|
| Minimum | 30' | 1.766 | | | 1.947 | | | 0.887 | | | 0.994 | | |
| TMP (℃) | 10' | 1.673 | 1.515 | 1.558 | 1.802 | 1.726 | 1.793 | 0.896 | 0.902 | 0.899 | 0.995 | 0.995 | 0.995 |
| | 5' | 1.338 | 1.292 | 1.325 | 1.666 | 1.503 | 1.582 | 0.904 | 0.937 | 0.923 | 0.995 | 0.995 | 0.995 |
| | 2.5' | 1.233 | 1.142 | 1.211 | 1.401 | 1.349 | 1.384 | 0.946 | 0.951 | 0.949 | 0.995 | 0.997 | 0.996 |
| | 0.5' | 1.140 | 1.050 | 1.137 | 1.322 | 1.248 | 1.271 | 0.955 | 0.972 | 0.963 | 0.997 | 0.998 | 0.997 |
| Mean | 30' | 1.598 | | | 1.759 | | | 0.888 | | | 0.996 | | |
| TMP (℃) | 10' | 1.277 | 1.140 | 1.188 | 1.433 | 1.293 | 1.358 | 0.899 | 0.914 | 0.904 | 0.997 | 0.997 | 0.997 |
| | 5' | 1.117 | 0.980 | 1.003 | 1.222 | 1.133 | 1.197 | 0.926 | 0.950 | 0.933 | 0.997 | 0.997 | 0.997 |
| | 2.5' | 0.977 | 0.836 | 0.859 | 1.157 | 0.988 | 0.993 | 0.966 | 0.976 | 0.973 | 0.997 | 0.998 | 0.997 |
| | 0.5' | 0.826 | 0.820 | 0.822 | 0.974 | 0.969 | 0.970 | 0.977 | 0.981 | 0.980 | 0.998 | 0.998 | 0.998 |
| Maximum | 30' | 2.034 | | | 2.206 | | | 0.800 | | | 0.995 | | |
| TMP (℃) | 10' | 1.800 | 1.672 | 1.755 | 2.044 | 1.886 | 1.968 | 0.811 | 0.832 | 0.824 | 0.995 | 0.996 | 0.996 |
| | 5' | 1.649 | 1.487 | 1.548 | 1.864 | 1.700 | 1.756 | 0.843 | 0.856 | 0.850 | 0.996 | 0.996 | 0.996 |
| | 2.5' | 1.455 | 1.310 | 1.387 | 1.666 | 1.523 | 1.632 | 0.875 | 0.909 | 0.887 | 0.996 | 0.997 | 0.996 |
| | 0.5' | 1.296 | 1.282 | 1.291 | 1.511 | 1.491 | 1.500 | 0.909 | 0.910 | 0.910 | 0.997 | 0.997 | 0.997 |
| PRE | 30' | 17.850 | | | 29.559 | | | 0.614 | | | 0.885 | | |
| (mm) | 10' | 16.884 | 16.647 | 16.741 | 28.022 | 27.559 | 27.946 | 0.675 | 0.735 | 0.700 | 0.887 | 0.890 | 0.890 |
| | 5' | 16.134 | 15.223 | 15.942 | 26.222 | 25.185 | 25.888 | 0.764 | 0.791 | 0.773 | 0.892 | 0.900 | 0.894 |
| | 2.5' | 14.867 | 14.024 | 14.557 | 24.374 | 23.191 | 23.867 | 0.791 | 0.792 | 0.791 | 0.914 | 0.920 | 0.919 |
| | 0.5' | 13.772 | 13.269 | 13.443 | 22.655 | 21.941 | 22.213 | 0.794 | 0.808 | 0.802 | 0.920 | 0.929 | 0.926 |

Notes: Res indicates the spatial resolution. The subscripts $c$, $l$, and $n$ indicate bicubic, bilinear, and nearest-neighbor interpolations, respectively. The original TMPs and PRE are the 30' CRU data and directly compared with the observed data. Evaluations at 10', 5', 2.5', and 0.5' are the evaluations for the downscaled datasets. MAE, RMSE, NSE, and Cor indicate the mean absolute error, root-mean-square error, Nash–Sutcliffe efficiency coefficient, and correlation coefficient.

The revision will add all the above sections to data description and results.

(3) As shown in Table 2, the uncertainty (values) are very large, which is much larger than the differences between observed and downscaled mean values. Therefore, it is very likely that the difference between observations and downscaled data is statistically insignificant unless the authors can prove that is indeed the case.

Response: The 'uncertainty' in Table 2 of the previous version were not the errors between observation and downscaled data, but the standard deviations of meteorological variables across the

745 weather stations. In fact, the evaluations for the downscaled data using time series indicated the observation and downscaled data matched well (above Table 3).

(4) As shown in Table 3, the authors focused on the statistical significance of the trends, but they ignored the more important question whether the differences among CRU, downscaled data, and observations are statistically significant. I suggest the authors using uncertainty (values) instead of "**" marking.

Response: Table 3 in the old version presented an indirect evaluation index by using change magnitudes of meteorological variables. The differences in change magnitudes between observation and downscaled datasets were smaller than those between observation and original data, indicating our technique improved data quality.

As suggested, in this version, we will analyze the correlation between original/downscaled CRU datasets and observations using the independent 496 stations (above Table 3). The correlation coefficients of downscaled vs observed data were greater than those between original CRU vs observation, implying that our newly generated data had good performance.

(5) Figure 4 and its discussion in the main text: Left and Right columns should be explained in the figure caption. I am wondering whether the correlations are mostly associated with climatologies. It should be more convincing if anomalies are used in the diagrams.

Response: The original figure 4 used average of the 0.5' downscaled data and observation. We think average or anomaly should show similar patterns since the anomalies are actually the current averages extracted by the observational averages. As your suggestion, we will analyze these based on the independent 496 stations in the revision.

Minor comments

P3L25, Delta downscaling, a reference is needed and a brief description is helpful.

Response: Thank you for the suggestion. We will expand the description of Delta downscaling procedure.

Delta downscaling was employed to generate monthly TMPs and PRE for the period 1901–2017 at spatial resolutions of 10', 5', 2.5', and 0.5'. The Delta downscaling procedure contains four steps (Peng et al., 2018).

The first step constructs the climatology for each month and each climatic variable based on the 30' CRU time series. In specific, the long-term average of TMPs and PRE were calculated for each month using CRU TMPs and PRE time series. This step keeps the spatial resolution of 30' from CRU. To match the data period of the WorldClim, the period 1970–2000 was selected.

The second calculates the anomaly time series for each climatic variable using the 30' CRU time series and the calculated monthly averages. The TMP anomaly was calculated as the difference between the TMP time series and their long-term average in each month, while the PRE anomaly was calculated as the ratio of the PRE time series to their long-term average in each month.

$$\text{An\_TMP}(yr, m) = \text{TMP}(yr, m) - \text{CRUClim\_TMP}(m) \qquad (1)$$

$$\text{An\_PRE}(yr, m) = \text{PRE}(yr, m) \ / \ \text{CRUClim\_PRE}(m) \qquad (2)$$

where An_TMP(*yr*, *m*) and An_PRE(*yr*, *m*) are the anomaly for temperatures and precipitation, respectively, at *m* month and *yr* year; TMP(*yr*, *m*) and PRE(*yr*, *m*) are the absolute temperatures and precipitation values, respectively, at *m* month and *yr* year; CRUClim_TMP(*m*) and CRUClim_PRE(*m*) are the 30' climatology for temperatures and precipitation, respectively, at *m* month. *m* ranges from 1 to 12, representing January to December.

The third step spatially interpolates the 30' anomaly time series to higher spatial resolution. In this study, the 30' anomaly was interpolated to four spatial resolutions (i.e., 10', 5', 2.5', and 0.5') to match those of reference datasets from the WorldClim. To optimize the interpolation, we compared the performance of three methods, including bicubic interpolation, bilinear interpolation, and nearest-neighbor interpolation methods, to select the appropriate approach.

The last step reversely transforms the time series of anomaly to those of absolute values. Contrary to the steps for anomaly calculation, addition was used for TMPs, while multiplication for PRE. It should be noted that this step was carried out for each high-spatial-resolution anomaly time series.

$$\text{TMP}(yr, m, res) = \text{An\_TMP}(yr, m, res) + \text{WorldClim\_TMP}(m, res) \qquad (3)$$

$$\text{PRE}(yr, m, res) = \text{An\_PRE}(yr, m, res) \times \text{WorldClim\_PRE}(m, res) \qquad (4)$$

where *res* represents the spatial resolution, i.e., 10', 5', 2.5', and 0.5'; TMP(*yr*, *m*, *res*) and PRE(*yr*, *m*, *res*) are the absolute temperatures and precipitation values with a spatial resolution of *res*, respectively, at *m* month and *yr* year; An_TMP(*yr*, *m*, *res*) and An_PRE(*yr*, *m*, *res*) are the anomaly with a spatial resolution of *res* for temperatures and precipitation, respectively, at *m* month and *yr* year; WorldClim_TMP(*m*, *res*) and WorldClim_PRE(*m*, *res*) are the climatology from the WorldClim with a spatial resolution of *res* for temperatures and precipitation, respectively, at *m* month.

Figure 2 presented the steps of Delta downscaling for mean TMP using the CRU 30' time series and WorldClim 0.5' climatology datasets.

[Figure]

Figure 2. Schematic illustration of the Delta downscaling procedure. The mean TMP (TMP_m) in July 2017 obtained from CRU is used as an example.

P3L28⡊30, The calculation of TMP anomaly is conventional, but why PRE anomaly is defined by ratio? What happens if the difference is defined for PRE?
Response: In the Delta downscaling procedure, difference for TMP and ratio for PRE are very conventional. Therefore, the so-called 'anomaly' here is actually not the traditional anomaly. If the traditional anomaly is calculated for PRE, the downscaled PRE may have negative values.

P4L10, NSE needs a reference.
Response: Thank you for this suggestion. We will add an reference in the revision, if we have this opportunity.

P4L16, "raw" data, CRU data can never be called "raw". How many station data are used in CRU over China? If all 745 station data are used in CRU, the comparison in Section 4.1 is not independent!
Response: We will replace the "raw CRU data" as the "original CRU data" in the revision. Observation from 323 weather stations across China were employed to generate the CRU data. For evaluation of downscaled dataset, the 323 weather stations will be excluded.

We appreciate for this comment on evaluation based on independent dataset. As shown above, we used additional 496 weather stations to evaluate the original CRU time series, WorldClim, and downscaled time series (above Figure 1 and Tables 1-3).

Further, we have analyzed the representativeness of the 496 independent stations over China region (Figure 3). Figure 3 shows the orographic statistic information (e.g., elevation, slope, and aspect) of China and the stations. The results presented that the proportions of the weather station numbers in different orographic gradients almost correspond to those in China excepting the areas with elevations exceeding 4500 m, which indicated that these weather stations could represent the climate variation over China and be used for validating the downscaled dataset. This exception is inevitable, because of the observability, installation, and maintenance of the weather stations in those areas. We will revised the related contents in the revision.

[Figure]

Figure 3. Orographic statistic information at different gradients for China and weather stations used in this study.

P5, Section 4.2, first paragraph, it is not clear whether the description is for the downscaled data of 0.5'. I also suggest use the same color scale for the TMPs in Figure 3. Second paragraph, see the major comments (2).
Response: Thanks for your suggestions. We will revise the manuscript accordingly.

P6L5, "downward" trend, check and verify it.
Response: In some grids (0.33 % of the land area of China), the 0.5' downscaled maximum TMP showed significant downward trends. This implies that the downscaled dataset could present the

spatial variability of climate variable with more details. However, due to data availability, we cannot verify it through comparing the results with observations.

Figure 5 (and Figure 3), the focus should be the difference between CRU and downscaled data rather than the trend (and climatology) itself.

Response: As we described in the method, the change magnitudes were used as an index for indirect evaluation. It can give additional information. In the new version, we will further enhance the evaluations of the original CRU and downscaled datasets.

---

## Author Comment (AC2) · 3 Aug 2019

This study by Peng et al. developed a high-resolution and long-term climate dataset over China. The CRU data was downscaled to 1km using the Delta downscaling framework. The topic is interesting, and the product would be useful in climate-related studies for the nation. However, I think the paper needs some improvement and further discussion before it can be published.

My major concerns include:

1.  The Delta downscaling improves the spatial representation of temperature/ precipitation climatology using high-resolution WorldClim as the reference climatology. However, it is hard to understand how this downscaling method improves temporal variability (or trend), because the temporal change is simply based on the interpolated anomalies from low-resolution CRU. This limitation should be at least explained and discussed in the manuscript.

Response: Thank you for this point. The Delta downscaling not only improved the spatial resolution of CRU time series, but also corrected the bias of CRU time series. Table 1 in the original manuscript clearly showed the good performances on bias correction.

Now, we have analyzed the correlations between original/downscaled and observed data in the time series (Table 1), based on the 496 independent stations (Figure 1), which were not taken part in the creation of the CRU time series and WorldClim climatology. The results indicated that downscaled datasets have better performance than the original CRU dataset, especially for the 0.5' dataset. Specifically, the bias has been improved very much, and the temporal variability has been slightly improved. These imply that the Delta downscaling used in this study can improve the temporal variability of original CRU time series. It should be attributed to the introduction of monthly WorldClim climatology for each climatic variable.

I understand your query very much. If one climatology was used as the reference climatology for the downscaling, the temporal variability of downscaled dataset is the same as the original dataset. However, this study employed 12 climatology layers, representing the climatology from Jan to Dec, to downscale the CRU data. Indeed, the description of downscaling method in the original manuscript is ambiguous, and we will revised this part in the revision.

Table 1. Statistical characteristics between original/downscaled and observed monthly TMPs and PRE in the time series (1951–2016). The values shown here are the averaged evaluation results at all 496 weather stations.

|  | Res | $MAE_c$ | $MAE_l$ | $MAE_n$ | $RMSE_c$ | $RMSE_l$ | $RMSE_n$ | $NSE_c$ | $NSE_l$ | $NSE_n$ | $Cor_c$ | $Cor_l$ | $Cor_n$ |
|---|---|---|---|---|---|---|---|---|---|---|---|---|---|
| Minimum | 30' | 1.766 | | | 1.947 | | | 0.887 | | | 0.994 | | |
| TMP (°C) | 10' | 1.673 | 1.515 | 1.558 | 1.802 | 1.726 | 1.793 | 0.896 | 0.902 | 0.899 | 0.995 | 0.995 | 0.995 |
|  | 5' | 1.338 | 1.292 | 1.325 | 1.666 | 1.503 | 1.582 | 0.904 | 0.937 | 0.923 | 0.995 | 0.995 | 0.995 |
|  | 2.5' | 1.233 | 1.142 | 1.211 | 1.401 | 1.349 | 1.384 | 0.946 | 0.951 | 0.949 | 0.995 | 0.997 | 0.996 |
|  | 0.5' | 1.140 | 1.050 | 1.137 | 1.322 | 1.248 | 1.271 | 0.955 | 0.972 | 0.963 | 0.997 | 0.998 | 0.997 |
| Mean | 30' | 1.598 | | | 1.759 | | | 0.888 | | | 0.996 | | |
| TMP (°C) | 10' | 1.277 | 1.140 | 1.188 | 1.433 | 1.293 | 1.358 | 0.899 | 0.914 | 0.904 | 0.997 | 0.997 | 0.997 |
|  | 5' | 1.117 | 0.980 | 1.003 | 1.222 | 1.133 | 1.197 | 0.926 | 0.950 | 0.933 | 0.997 | 0.997 | 0.997 |
|  | 2.5' | 0.977 | 0.836 | 0.859 | 1.157 | 0.988 | 0.993 | 0.966 | 0.976 | 0.973 | 0.997 | 0.998 | 0.997 |
|  | 0.5' | 0.826 | 0.820 | 0.822 | 0.974 | 0.969 | 0.970 | 0.977 | 0.981 | 0.980 | 0.998 | 0.998 | 0.998 |
| Maximum | 30' | 2.034 | | | 2.206 | | | 0.800 | | | 0.995 | | |
| TMP (°C) | 10' | 1.800 | 1.672 | 1.755 | 2.044 | 1.886 | 1.968 | 0.811 | 0.832 | 0.824 | 0.995 | 0.996 | 0.996 |

| | Res | MAE$_c$ | MAE$_l$ | MAE$_n$ | RMSE$_c$ | RMSE$_l$ | RMSE$_n$ | NSE$_c$ | NSE$_l$ | NSE$_n$ | Cor$_c$ | Cor$_l$ | Cor$_n$ |
|---|---|---|---|---|---|---|---|---|---|---|---|---|---|
| | 5' | 1.649 | 1.487 | 1.548 | 1.864 | 1.700 | 1.756 | 0.843 | 0.856 | 0.850 | 0.996 | 0.996 | 0.996 |
| | 2.5' | 1.455 | 1.310 | 1.387 | 1.666 | 1.523 | 1.632 | 0.875 | 0.909 | 0.887 | 0.996 | 0.997 | 0.996 |
| | 0.5' | 1.296 | 1.282 | 1.291 | 1.511 | 1.491 | 1.500 | 0.909 | 0.910 | 0.910 | 0.997 | 0.997 | 0.997 |
| PRE | 30' | 17.850 | | | 29.559 | | | 0.614 | | | 0.885 | | |
| (mm) | 10' | 16.884 | 16.647 | 16.741 | 28.022 | 27.559 | 27.946 | 0.675 | 0.735 | 0.700 | 0.887 | 0.890 | 0.890 |
| | 5' | 16.134 | 15.223 | 15.942 | 26.222 | 25.185 | 25.888 | 0.764 | 0.791 | 0.773 | 0.892 | 0.900 | 0.894 |
| | 2.5' | 14.867 | 14.024 | 14.557 | 24.374 | 23.191 | 23.867 | 0.791 | 0.792 | 0.791 | 0.914 | 0.920 | 0.919 |
| | 0.5' | 13.772 | 13.269 | 13.443 | 22.655 | 21.941 | 22.213 | 0.794 | 0.808 | 0.802 | 0.920 | 0.929 | 0.926 |

Notes: Res indicates the spatial resolution. The subscripts *c*, *l*, and *n* indicate bicubic, bilinear, and nearest-neighbor interpolations, respectively. The original TMPs and PRE are the 30' CRU data and directly compared with the observed data. Evaluations at 10', 5', 2.5', and 0.5' are the evaluations for the downscaled datasets. MAE, RMSE, NSE, and Cor indicate the mean absolute error, root-mean-square error, Nash–Sutcliffe efficiency coefficient, and correlation coefficient.

[Figure]

DEM (m)
8844
-154

0       500      1,000 km

▲ Stations for producing the CRU TS and WorldClim data (323 )

• Stations for validating the downscaled results in this study (496)

N

Figure 1. Spatial distribution of the national weather stations across China.

2. The downscaled dataset is developed based on the WorldClim reference climatology from 1970 to 2000, and the data evaluation is performed for the period 1951-2016. Can the authors first evaluate the reference data (WorldClim) at different resolutions? Also, because there is an overlapping period for data training and evaluation, is it possible to use two separate periods, in which one is for downscaling and the other one is for data evaluation?

Response: Thanks for the suggestion. Your proposal of two separate periods (one period for downscaling and the other period for evaluation) was substituted by another method.

Some of the weather stations used in our original manuscript have been involved in the data

assimilation of CRU and WorldClim data. They were also used for the data evaluation. Now, we kicked out those stations and evaluated the WorldClim, CRU and downscaled data only with 496 stations (above Figure 1). The modified procedure can improve the reliability of the evaluation (Tables 1-3). Overall, the WorldClim datasets have high performance to represent the monthly climatology over China region, and the dataset performs better for higher spatial resolution. In specific, the absolute errors become smaller with increasing spatial resolution (Table 2) and the correlations get greater with increasing spatial resolution (Table 3).

Table 2. The mean absolute errors between the observed and WorldClim climatology at different spatial resolutions over the 496 weather stations. The period ranges from 1970 to 2000.

|  |  | Jan | Feb | Mar | Apr | May | Jun | Jul | Aug | Sep | Oct | Nov | Dec |
|---|---|---|---|---|---|---|---|---|---|---|---|---|---|
| Minimum | 10' | 0.726 | 0.675 | 0.615 | 0.533 | 0.515 | 0.533 | 0.789 | 0.759 | 0.719 | 0.639 | 0.643 | 0.656 |
| TMP (°C) | 5' | 0.653 | 0.596 | 0.521 | 0.467 | 0.450 | 0.429 | 0.660 | 0.633 | 0.607 | 0.523 | 0.514 | 0.550 |
|  | 2.5' | 0.632 | 0.563 | 0.484 | 0.433 | 0.411 | 0.372 | 0.602 | 0.574 | 0.543 | 0.459 | 0.449 | 0.503 |
|  | 0.5' | 0.622 | 0.549 | 0.474 | 0.430 | 0.408 | 0.354 | 0.567 | 0.541 | 0.513 | 0.428 | 0.420 | 0.484 |
|  |  |  |  |  |  |  |  |  |  |  |  |  |  |
| Mean | 10' | 0.450 | 0.481 | 0.470 | 0.482 | 0.487 | 0.478 | 0.455 | 0.445 | 0.427 | 0.425 | 0.425 | 0.427 |
| TMP (°C) | 5' | 0.401 | 0.426 | 0.385 | 0.390 | 0.400 | 0.391 | 0.379 | 0.387 | 0.380 | 0.367 | 0.362 | 0.377 |
|  | 2.5' | 0.365 | 0.378 | 0.338 | 0.332 | 0.351 | 0.342 | 0.338 | 0.356 | 0.348 | 0.333 | 0.331 | 0.349 |
|  | 0.5' | 0.355 | 0.366 | 0.328 | 0.322 | 0.337 | 0.330 | 0.334 | 0.351 | 0.343 | 0.331 | 0.324 | 0.342 |
|  |  |  |  |  |  |  |  |  |  |  |  |  |  |
| Maximum | 10' | 0.832 | 0.821 | 0.809 | 0.909 | 0.827 | 0.678 | 0.718 | 0.734 | 0.644 | 0.658 | 0.630 | 0.687 |
| TMP (°C) | 5' | 0.727 | 0.711 | 0.666 | 0.760 | 0.687 | 0.560 | 0.645 | 0.658 | 0.568 | 0.561 | 0.511 | 0.576 |
|  | 2.5' | 0.664 | 0.637 | 0.591 | 0.670 | 0.597 | 0.485 | 0.589 | 0.600 | 0.531 | 0.509 | 0.447 | 0.517 |
|  | 0.5' | 0.631 | 0.596 | 0.544 | 0.611 | 0.544 | 0.445 | 0.574 | 0.578 | 0.516 | 0.484 | 0.405 | 0.479 |
|  |  |  |  |  |  |  |  |  |  |  |  |  |  |
| PRE | 10' | 2.165 | 1.869 | 3.476 | 4.662 | 5.651 | 8.416 | 9.716 | 7.993 | 5.825 | 3.968 | 2.202 | 1.378 |
| (mm) | 5' | 2.077 | 1.834 | 3.407 | 4.641 | 5.637 | 8.291 | 9.702 | 7.841 | 5.805 | 3.908 | 2.183 | 1.348 |
|  | 2.5' | 2.074 | 1.813 | 3.404 | 4.603 | 5.594 | 8.268 | 9.664 | 7.705 | 5.742 | 3.904 | 2.182 | 1.334 |
|  | 0.5' | 2.072 | 1.797 | 3.360 | 4.495 | 5.564 | 8.190 | 9.630 | 7.651 | 5.699 | 3.895 | 2.170 | 1.300 |

Table 3. The correlation coefficients between the observed and WorldClim climatology at different spatial resolutions over the 496 weather stations. The period ranges from 1970 to 2000.

|  |  | Jan | Feb | Mar | Apr | May | Jun | Jul | Aug | Sep | Oct | Nov | Dec |
|---|---|---|---|---|---|---|---|---|---|---|---|---|---|
| Minimum | 10' | 0.987 | 0.984 | 0.977 | 0.969 | 0.963 | 0.962 | 0.955 | 0.957 | 0.956 | 0.971 | 0.984 | 0.987 |
| TMP (°C) | 5' | 0.989 | 0.987 | 0.983 | 0.977 | 0.973 | 0.973 | 0.964 | 0.966 | 0.968 | 0.980 | 0.990 | 0.991 |
|  | 2.5' | 0.989 | 0.988 | 0.985 | 0.981 | 0.978 | 0.977 | 0.968 | 0.971 | 0.974 | 0.985 | 0.992 | 0.992 |
|  | 0.5' | 0.989 | 0.989 | 0.986 | 0.983 | 0.981 | 0.980 | 0.972 | 0.974 | 0.977 | 0.988 | 0.993 | 0.993 |
|  |  |  |  |  |  |  |  |  |  |  |  |  |  |
| Mean | 10' | 0.986 | 0.979 | 0.968 | 0.955 | 0.949 | 0.949 | 0.956 | 0.958 | 0.966 | 0.974 | 0.982 | 0.987 |
| TMP (°C) | 5' | 0.991 | 0.986 | 0.980 | 0.969 | 0.962 | 0.959 | 0.963 | 0.965 | 0.973 | 0.983 | 0.989 | 0.991 |
|  | 2.5' | 0.993 | 0.990 | 0.986 | 0.977 | 0.970 | 0.965 | 0.968 | 0.970 | 0.978 | 0.986 | 0.992 | 0.993 |
|  | 0.5' | 0.994 | 0.992 | 0.989 | 0.981 | 0.973 | 0.968 | 0.970 | 0.972 | 0.980 | 0.988 | 0.993 | 0.995 |

| | | | | | | | | | | | | | |
|---|---|---|---|---|---|---|---|---|---|---|---|---|---|
| Maximum | 10' | 0.958 | 0.946 | 0.920 | 0.892 | 0.889 | 0.899 | 0.893 | 0.890 | 0.935 | 0.957 | 0.968 | 0.974 |
| TMP (°C) | 5' | 0.969 | 0.961 | 0.946 | 0.921 | 0.912 | 0.912 | 0.898 | 0.896 | 0.939 | 0.965 | 0.978 | 0.982 |
| | 2.5' | 0.976 | 0.971 | 0.960 | 0.941 | 0.930 | 0.925 | 0.910 | 0.909 | 0.945 | 0.971 | 0.984 | 0.986 |
| | 0.5' | 0.979 | 0.976 | 0.968 | 0.951 | 0.940 | 0.932 | 0.913 | 0.912 | 0.946 | 0.973 | 0.988 | 0.989 |
| | | | | | | | | | | | | | |
| PRE | 10' | 0.976 | 0.980 | 0.978 | 0.979 | 0.974 | 0.961 | 0.903 | 0.920 | 0.941 | 0.908 | 0.939 | 0.965 |
| (mm) | 5' | 0.976 | 0.980 | 0.979 | 0.979 | 0.974 | 0.961 | 0.905 | 0.924 | 0.943 | 0.911 | 0.940 | 0.966 |
| | 2.5' | 0.976 | 0.981 | 0.980 | 0.979 | 0.974 | 0.962 | 0.908 | 0.930 | 0.943 | 0.913 | 0.941 | 0.967 |
| | 0.5' | 0.977 | 0.981 | 0.981 | 0.980 | 0.975 | 0.962 | 0.909 | 0.930 | 0.944 | 0.914 | 0.941 | 0.968 |

3. The authors need to discuss the possible reasons why CRU temperatures have systematic cold biases.

Response: The CRU group introduced the averaged 30' DEM to generate the CRU data, which weakens the representation of temperatures in the actual land surface, especially in the regions with complex terrain. Moreover, the evaluation for the original CRU data was carried out at the station scale. For instance, it may present a warm bias if the averaged grid elevation involved by the CRU data is lower than the station elevation, while a cold bias if the averaged grid elevation involved by the CRU data is greater than the station elevation. The weather stations used in this study belong to the national weather station, which were often established in the valley near to the county or city. Thus, it is very likely that the averaged grid elevation involved by the CRU data is greater than the station elevation at most of the weather station, and presenting the "cold bias".

In the revision, we will enhance the related discussion. In the Data section, we will add additional information to describe how CRU generated their data as follows.

Methodologies used by CRU group to construct 30' time series dataset are similar to the Delta downscaling framework employed herein. First, more than 5000 weather stations are employed, and each station series is converted to anomalies by subtracting (for temperatures) or dividing (for precipitation) the 1961–1990 normal from all that station's data. Then, the station anomaly time series are linearly interpolated into 30' grids covering the global land surface. Finally, the grid anomaly time series were transformed back to absolute monthly values by a 30' climatology during 1961–1990. Specifically, the 30' climatology used by the CRU group contain the climatology for each month and are obtained from New et al. (1999). This climatology were generated by a function considering the latitude, longitude, and elevation, base on global 3615–19800 weather stations. Elevation data used in this climatology had a spatial resolution of 30', which was a mean result of global 5' digital elevation model. Specifically, elevation at each 30' grid was the mean of 36 grids of 5' digital elevation model (New et al., 1999). Therefore, the CRU dataset could well represent orographic effects on climate variation at the 30' spatial resolution; compared with similar gridded products, the CRU dataset had better performance. In addition, 323 weather stations across China region were employed by CRU group to generate the CRU time series data (Harris et al., 2014) (above Figure 1).

4. The dataset covers from 1901 to 2017, but most of the evaluations and discussion are about post-1950. Data quality or uncertainties before 1950 need more discussion.

Response: Although China has some weather stations with data starting from 1901, all of them have

been used to generate the CRU time series. We thus cannot verify the data quality before 1950 because of data availability. However, our downscaling procedure used data from CRU and WordClim and did not incorporate the observation, the data quality thus mainly depends on that of the data center. Our evaluation showed that the data quality of the data centers is overall satisfactory, and our downscaling procedure can further improve the data quality.

Specific Comments:

1. P3, L10-L15: Can we get the information about how many stations in China were used for CRU TS and WorldClim? How different are they? Are they comparable to the 745 weather stations used in this study?

Response: There have several overlapped weather stations for the creations of CRU and WorldClim data as well as the evaluation in the original manuscript. 323 weather stations across China region (above Figure 1) were employed by CRU group to generate the CRU time series data (Harris et al., 2014), and the WorldClim monthly datasets were generated using all of these stations obtained from the CRU group (Fick and Hijmans, 2017). For the independent evaluations in this study, we have used the 496 independent weather stations to evaluate the original CRU time series, WorldClim, and downscaled time series (above Figure 1 and Tables 1-3). We will revised the related contents in the revision.

Further, we have analyzed the representativeness of the 496 independent stations over China region (Figure 2). Figure 2 shows the orographic statistic information (e.g., elevation, slope, and aspect) of China and the stations. The results presented that the proportions of the weather station numbers in different orographic gradients almost correspond to those in China excepting the areas with elevations exceeding 4500 m, which indicated that these weather stations could represent the climate variation over China and be used for validating the downscaled dataset. This exception is inevitable, because of the observability, installation, and maintenance of the weather stations in those areas. We will revised the related contents in the revision.

[Figure]

Figure 2. Orographic statistic information at different gradients for China and weather stations used in this study.

2. P4, L17-19: I assume the final product is generated using the bilinear interpolation method? This should be mentioned in conclusion and abstract as well.

Response: Yes. The bilinear method was finally used to interpolation. We will add it in the revision.

3. P5, section 4.3: Trend is one aspect of the temporal variations. It would be better to also calculate the correlation of the time series.

Response: Thank you for this suggestion. We calculated the correlation and presented the results in Table 1.

4. Table 1: These metrics are applied to the climatology of TMP and PRE for 1951-2016? Or applied to time series of monthly TMP and PRE, then averaged over the 745 stations? Or any other way? This should be clarified in the main text or table caption. Same issue for Table S1.

Response: These metrics are applied to time series of monthly TMP and PRE, then averaged over the 745 stations. Based on the 496 independent stations, we have calculated the results, as well as the correlation coefficient of time series (above Table 1). We will revise the related contents as your suggestions.

5. Figure 4: Because the climatology is "corrected" using high-resolution reference data, it is not surprising that the downscaled data outperformed the CRU data in terms of the climatology. As suggested in Comment #3, it would be better to have a similar figure to show the time series (or anomalies).

Response: We will revise the related contents as your suggestions, based on the independent stations.

6. Figures 6-9: These figures are not quite informative. It would be better to add the trends as text on the figures. For figure 9, it is really hard to distinguish those three lines.

Response: We will revise the related contents as your suggestions, based on the independent stations.

---

## Author Comment (AC3) · 3 Aug 2019

The authors proposed a high-spatial-resolution monthly temperature and precipitation dataset for China by Delta downscaling of CRU dataset. The original CRU at 30' resolution is downscaled to 1km grid. The new downscaled data set include four common climate elements that are always the driven data for various models. This topic is quite interesting and would be useful for the climate change community. However, there are some obvious flaws in the downscaling procedure and the evaluation part. More interpretation and discussion should be improved. Therefore, I do not support this publication in ESSD at current version.

General comments:

1. Downscaling is a complicated procedure, especially for precipitation from 30' to 1km grid. I do not agree that the downscaled data set represents the local physical process. Actually, Delta downscaling is an interpolation method. CRU data set is also actually produced by interpolation method. The final downscaling result is the sum of "raw" CRU and interpolated anomaly. For my understanding, there is not any physical process involved. Conventionally, for a better local representation, local topography features should be considered such as aspect, slope and elevation.

Response: You are right. The methods that CRU used to construct 30' time series dataset are similar to the Delta downscaling framework. However, this study further improved the CRU data to higher spatial resolution by combining CRU and WorldClim datasets. The WorldClim dataset have four spatial resolutions (i.e., 10', 5', 2.5', and 0.5'), which consider the effects of local topography features, distance to the nearest coast, and three satellite-derived covariates. Thus, the downscaled datasets considered the above physical process. Our manuscript did not present this detailed information. We will add them to the revised version.

For the CRU time series data, we will describe it as follows.

Monthly mean, maximum, and minimum TMPs, as well as PRE, with a spatial resolution of 30' and covering a period from January 1901 to December 2017, were obtained from the CRU TS v. 4.02 dataset (http://www.cru.uea.ac.uk) (Harris et al., 2014). Methodologies used by CRU group to construct 30' time series dataset are similar to the Delta downscaling framework. First, more than 5000 weather stations are employed, and each station series is converted to anomalies by subtracting (for temperatures) or dividing (for precipitation) the 1961–1990 normal from all that station's data. Then, the station anomaly time series are linearly interpolated into 30' grids covering the global land surface. Finally, the grid anomaly time series were transformed back to absolute monthly values by a 30' climatology during 1961–1990. Specifically, the 30' climatology used by the CRU group contain the climatology for each month and are obtained from New et al. (1999). This climatology were generated by a function considering the latitude, longitude, and elevation, base on global 3615–19800 weather stations. Elevation data used in this climatology had a spatial resolution of 30', which was a mean result of global 5' digital elevation model. Specifically, elevation at each 30' grid was the mean of 36 grids of 5' digital elevation model (New et al., 1999). Therefore, the CRU dataset could well represent orographic effects on climate variation at the 30' spatial resolution; compared with similar gridded products, the CRU dataset had better performance. In addition, 323 weather stations across China region were employed by CRU group to generate the CRU time series data (Harris et al., 2014) (Figure 1).

[Figure]

▲  Stations for producing the CRU TS and WorldClim data (323 )

•  Stations for validating the downscaled results in this study (496)

Figure 1. Spatial distribution of the national weather stations across China.

For the WorldClim data, we will describe it as follows.

To downscale CRU TMPs and PRE time series to higher spatial resolutions, we obtained four high-resolution reference datasets at spatial resolutions of 10', 5', 2.5', and 0.5' from the WorldClim v. 2.0 (http://worldclim.org) (Fick and Hijmans, 2017). The reference datasets contained monthly averages of climatic variables (mean, maximum, and minimum TMPs, as well as PRE) for the period 1970–2000, generated based on global 9000–60000 weather stations, using the thin-plate splines interpolation method. Thus, for each climatic variable, it has 12 climatology layers, representing the climatology ranging from January to December. Remarkably, the interpolation considered covariation with the latitude, longitude, elevation, distance to the nearest coast, and three satellite-derived covariates: the maximum and minimum land surface temperature and cloud cover, obtained from the MODIS satellite platform. Thus, these reference data reflect orographic effects and observed climate information for each month. Moreover, cross-validation correlations indicated that these reference data held good performance around the world, because of the introduction of satellite-derived and distance to the nearest coast covariates. In addition, weather stations over China region used in the WorldClim were the same as the CRU group (Fick and Hijmans, 2017) (Figure 1). For independent evaluation of downscaled dataset in this study, these weather stations would be excluded.

For the Delta downscaling method, we will describe it as follows.

Delta downscaling was employed to generate monthly TMPs and PRE for the period 1901–2017 at spatial resolutions of 10', 5', 2.5', and 0.5'. The Delta downscaling procedure contains four steps (Peng et al., 2018).

The first step constructs the climatology for each month and each climatic variable based on the 30'
CRU time series. In specific, the long-term average of TMPs and PRE were calculated for each
month using CRU TMPs and PRE time series. This step keeps the spatial resolution of 30' from
CRU. To match the data period of the WorldClim, the period 1970–2000 was selected.

The second calculates the anomaly time series for each climatic variable using the 30' CRU time
series and the calculated monthly averages. The TMP anomaly was calculated as the difference
between the TMP time series and their long-term average in each month, while the PRE anomaly
was calculated as the ratio of the PRE time series to their long-term average in each month.

$$\text{An\_TMP}(yr, m) = \text{TMP}(yr, m) - \text{CRUClim\_TMP}(m) \qquad (1)$$

$$\text{An\_PRE}(yr, m) = \text{PRE}(yr, m) \, / \, \text{CRUClim\_PRE}(m) \qquad (2)$$

where An_TMP($yr$, $m$) and An_PRE($yr$, $m$) are the anomaly for temperatures and precipitation,
respectively, at $m$ month and $yr$ year; TMP($yr$, $m$) and PRE($yr$, $m$) are the absolute temperatures and
precipitation values, respectively, at $m$ month and $yr$ year; CRUClim_TMP($m$) and
CRUClim_PRE($m$) are the 30' climatology for temperatures and precipitation, respectively, at $m$
month. $m$ ranges from 1 to 12, representing January to December.

The third step spatially interpolates the 30' anomaly time series to higher spatial resolution. In this
study, the 30' anomaly was interpolated to four spatial resolutions (i.e., 10', 5', 2.5', and 0.5') to
match those of reference datasets from the WorldClim. To optimize the interpolation, we compared
the performance of three methods, including bicubic interpolation, bilinear interpolation, and
nearest-neighbor interpolation methods, to select the appropriate approach.

The last step reversely transforms the time series of anomaly to those of absolute values. Contrary
to the steps for anomaly calculation, addition was used for TMPs, while multiplication for PRE. It
should be noted that this step was carried out for each high-spatial-resolution anomaly time series.

$$\text{TMP}(yr, m, res) = \text{An\_TMP}(yr, m, res) + \text{WorldClim\_TMP}(m, res) \qquad (3)$$

$$\text{PRE}(yr, m, res) = \text{An\_PRE}(yr, m, res) \times \text{WorldClim\_PRE}(m, res) \qquad (4)$$

where $res$ represents the spatial resolution, i.e., 10', 5', 2.5', and 0.5'; TMP($yr$, $m$, $res$) and PRE($yr$,
$m$, $res$) are the absolute temperatures and precipitation values with a spatial resolution of $res$,
respectively, at $m$ month and $yr$ year; An_TMP($yr$, $m$, $res$) and An_PRE($yr$, $m$, $res$) are the anomaly
with a spatial resolution of $res$ for temperatures and precipitation, respectively, at $m$ month and $yr$
year; WorldClim_TMP($m$, $res$) and WorldClim_PRE($m$, $res$) are the climatology from the
WorldClim with a spatial resolution of $res$ for temperatures and precipitation, respectively, at $m$
month.

Figure 2 presented the steps of Delta downscaling for mean TMP using the CRU 30' time series and
WorldClim 0.5' climatology datasets.

[Figure]

Figure 2. Schematic illustration of the Delta downscaling procedure. The mean TMP (TMP_m) in July 2017 obtained from CRU is used as an example.

2.  WorldClim data set is used as the reference data in downscaling. However, how well does WorldClim represent the climatology over China? I did not find this information in the current version. The bias of WorldClim could be transferred into the final results. Therefore, it is not easy to understand why the downscaled data has a better performance. If the authors use other reference data, how will the downscaling result be?

Response: There have several overlapped weather stations for the creations of CRU and WorldClim data as well as the evaluation in the original manuscript. Now, based on the independent stations, we evaluate the monthly WorldClim data for each climatic variables (Tables 1-2). The evaluation indicates that (1) the WorldClim datasets have high performance to represent the monthly climatology over China region; and (2) the performance is better almost along the higher spatial resolution. Therefore, the accuracy representation of monthly WorldClim data result in the better performance of the downscaled dataset, especially at the spatial resolution of 0.5'. Considering the advantage of the monthly WorldClim data (e.g., the introduction of satellite-derived and distance to the nearest coast covariates), we just used this reference data. Besides, there are no other reference data with different resolutions for this study. We will added above evaluation in the revision.

Table 1. The mean absolute errors between the observed and WorldClim climatology at different

spatial resolutions over the 496 weather stations. The period ranges from 1970 to 2000.

| | | Jan | Feb | Mar | Apr | May | Jun | Jul | Aug | Sep | Oct | Nov | Dec |
|---|---|---|---|---|---|---|---|---|---|---|---|---|---|
| Minimum | 10' | 0.726 | 0.675 | 0.615 | 0.533 | 0.515 | 0.533 | 0.789 | 0.759 | 0.719 | 0.639 | 0.643 | 0.656 |
| TMP (°C) | 5' | 0.653 | 0.596 | 0.521 | 0.467 | 0.450 | 0.429 | 0.660 | 0.633 | 0.607 | 0.523 | 0.514 | 0.550 |
| | 2.5' | 0.632 | 0.563 | 0.484 | 0.433 | 0.411 | 0.372 | 0.602 | 0.574 | 0.543 | 0.459 | 0.449 | 0.503 |
| | 0.5' | 0.622 | 0.549 | 0.474 | 0.430 | 0.408 | 0.354 | 0.567 | 0.541 | 0.513 | 0.428 | 0.420 | 0.484 |
| | | | | | | | | | | | | | |
| Mean | 10' | 0.450 | 0.481 | 0.470 | 0.482 | 0.487 | 0.478 | 0.455 | 0.445 | 0.427 | 0.425 | 0.425 | 0.427 |
| TMP (°C) | 5' | 0.401 | 0.426 | 0.385 | 0.390 | 0.400 | 0.391 | 0.379 | 0.387 | 0.380 | 0.367 | 0.362 | 0.377 |
| | 2.5' | 0.365 | 0.378 | 0.338 | 0.332 | 0.351 | 0.342 | 0.338 | 0.356 | 0.348 | 0.333 | 0.331 | 0.349 |
| | 0.5' | 0.355 | 0.366 | 0.328 | 0.322 | 0.337 | 0.330 | 0.334 | 0.351 | 0.343 | 0.331 | 0.324 | 0.342 |
| | | | | | | | | | | | | | |
| Maximum | 10' | 0.832 | 0.821 | 0.809 | 0.909 | 0.827 | 0.678 | 0.718 | 0.734 | 0.644 | 0.658 | 0.630 | 0.687 |
| TMP (°C) | 5' | 0.727 | 0.711 | 0.666 | 0.760 | 0.687 | 0.560 | 0.645 | 0.658 | 0.568 | 0.561 | 0.511 | 0.576 |
| | 2.5' | 0.664 | 0.637 | 0.591 | 0.670 | 0.597 | 0.485 | 0.589 | 0.600 | 0.531 | 0.509 | 0.447 | 0.517 |
| | 0.5' | 0.631 | 0.596 | 0.544 | 0.611 | 0.544 | 0.445 | 0.574 | 0.578 | 0.516 | 0.484 | 0.405 | 0.479 |
| | | | | | | | | | | | | | |
| PRE | 10' | 2.165 | 1.869 | 3.476 | 4.662 | 5.651 | 8.416 | 9.716 | 7.993 | 5.825 | 3.968 | 2.202 | 1.378 |
| (mm) | 5' | 2.077 | 1.834 | 3.407 | 4.641 | 5.637 | 8.291 | 9.702 | 7.841 | 5.805 | 3.908 | 2.183 | 1.348 |
| | 2.5' | 2.074 | 1.813 | 3.404 | 4.603 | 5.594 | 8.268 | 9.664 | 7.705 | 5.742 | 3.904 | 2.182 | 1.334 |
| | 0.5' | 2.072 | 1.797 | 3.360 | 4.495 | 5.564 | 8.190 | 9.630 | 7.651 | 5.699 | 3.895 | 2.170 | 1.300 |

Table 2. The correlation coefficients between the observed and WorldClim climatology at different spatial resolutions over the 496 weather stations. The period ranges from 1970 to 2000.

| | | Jan | Feb | Mar | Apr | May | Jun | Jul | Aug | Sep | Oct | Nov | Dec |
|---|---|---|---|---|---|---|---|---|---|---|---|---|---|
| Minimum | 10' | 0.987 | 0.984 | 0.977 | 0.969 | 0.963 | 0.962 | 0.955 | 0.957 | 0.956 | 0.971 | 0.984 | 0.987 |
| TMP (°C) | 5' | 0.989 | 0.987 | 0.983 | 0.977 | 0.973 | 0.973 | 0.964 | 0.966 | 0.968 | 0.980 | 0.990 | 0.991 |
| | 2.5' | 0.989 | 0.988 | 0.985 | 0.981 | 0.978 | 0.977 | 0.968 | 0.971 | 0.974 | 0.985 | 0.992 | 0.992 |
| | 0.5' | 0.989 | 0.989 | 0.986 | 0.983 | 0.981 | 0.980 | 0.972 | 0.974 | 0.977 | 0.988 | 0.993 | 0.993 |
| | | | | | | | | | | | | | |
| Mean | 10' | 0.986 | 0.979 | 0.968 | 0.955 | 0.949 | 0.949 | 0.956 | 0.958 | 0.966 | 0.974 | 0.982 | 0.987 |
| TMP (°C) | 5' | 0.991 | 0.986 | 0.980 | 0.969 | 0.962 | 0.959 | 0.963 | 0.965 | 0.973 | 0.983 | 0.989 | 0.991 |
| | 2.5' | 0.993 | 0.990 | 0.986 | 0.977 | 0.970 | 0.965 | 0.968 | 0.970 | 0.978 | 0.986 | 0.992 | 0.993 |
| | 0.5' | 0.994 | 0.992 | 0.989 | 0.981 | 0.973 | 0.968 | 0.970 | 0.972 | 0.980 | 0.988 | 0.993 | 0.995 |
| | | | | | | | | | | | | | |
| Maximum | 10' | 0.958 | 0.946 | 0.920 | 0.892 | 0.889 | 0.899 | 0.893 | 0.890 | 0.935 | 0.957 | 0.968 | 0.974 |
| TMP (°C) | 5' | 0.969 | 0.961 | 0.946 | 0.921 | 0.912 | 0.912 | 0.898 | 0.896 | 0.939 | 0.965 | 0.978 | 0.982 |
| | 2.5' | 0.976 | 0.971 | 0.960 | 0.941 | 0.930 | 0.925 | 0.910 | 0.909 | 0.945 | 0.971 | 0.984 | 0.986 |
| | 0.5' | 0.979 | 0.976 | 0.968 | 0.951 | 0.940 | 0.932 | 0.913 | 0.912 | 0.946 | 0.973 | 0.988 | 0.989 |
| | | | | | | | | | | | | | |
| PRE | 10' | 0.976 | 0.980 | 0.978 | 0.979 | 0.974 | 0.961 | 0.903 | 0.920 | 0.941 | 0.908 | 0.939 | 0.965 |
| (mm) | 5' | 0.976 | 0.980 | 0.979 | 0.979 | 0.974 | 0.961 | 0.905 | 0.924 | 0.943 | 0.911 | 0.940 | 0.966 |
| | 2.5' | 0.976 | 0.981 | 0.980 | 0.979 | 0.974 | 0.962 | 0.908 | 0.930 | 0.943 | 0.913 | 0.941 | 0.967 |
| | 0.5' | 0.977 | 0.981 | 0.981 | 0.980 | 0.975 | 0.962 | 0.909 | 0.930 | 0.944 | 0.914 | 0.941 | 0.968 |

3.  In addition, the "Direct evaluation" is not adequate. The time series are different for CRU, WorldClim, and observation. How do the authors guarantee the consistency of time series, in particular the period 1901-1950? Meanwhile, the mean climatology is calculated from 1970-2000. Is this time period appropriate for representation? For precipitation, the observation has shown significant nonstationary features after 1980s in China under the global warming. Unfortunately, Delta downscaling method does not consider the nonstationary.

Response: We will revised the "Direct evaluation" in the revision.

The Delta downscaling method involved the monthly reference climatology during a period. It is necessary to guarantee the accuracy representation of monthly reference climatology during this period. However, it is not necessary to guarantee the consistency of time series between WorldClim and other datasets. The details on the application of WorldClim data could be found in the responses on General comment #1. Now, we have evaluated the monthly WorldClim data for each climatic variables (above Tables 1-2), and the WorldClim data can represent the monthly climatology over China region for each climatic variable.

Indeed, the Delta downscaling method does not consider the "nonstationary", while it has the ability to enhance the spatial resolution and reliability for the 30' CRU time series. Thus, whether the precipitation over China has significant nonstationary features after 1980s mainly depends on the original CRU time series.

4.  The authors evaluated the new data set using 745 observations over China. I think it is not enough, especially for the west of China, such as the high mountains areas and Tibet Plateau. Meanwhile, most observations begin after 1950, how about the pre-1950? Therefore, it is hard to conclude the data set is "sufficiently reliable".

Response: Many thanks for your queries. For the independent evaluation in this study, we will use 496 independent weather stations (above Figure 1) to evaluate the original CRU time series, WorldClim, and downscaled time series. Further, we have analyzed the representativeness of the 496 independent stations over China region (Figure 3). Figure 3 shows the orographic statistic information (e.g., elevation, slope, and aspect) of China and the stations. The results presented that the proportions of the weather station numbers in different orographic gradients almost correspond to those in China excepting the areas with elevations exceeding 4500 m, which indicated that these weather stations could represent the climate variation over China and be used for validating the downscaled dataset. This exception is inevitable, because of the observability, installation, and maintenance of the weather stations in those areas. We will revised the related contents in the revision.

Although China has some weather stations with data starting from 1901, all of them have been used to generate the CRU time series. We thus cannot verify the data quality before 1950 because of data availability. However, our downscaling procedure used data from CRU and WordClim and did not incorporate the observation, the data quality thus mainly depends on that of the data center. Our evaluation showed that the data quality of the data centers is overall satisfactory, and our downscaling procedure can further improve the data quality. We will revised the related contents in the revision.

[Figure]

Figure 3. Orographic statistic information at different gradients for China and weather stations used in this study.

5. How many observations have been used in CRU and WorldClim? These sites should be excluded since they destroy the independence of evaluation.

Response: 323 weather stations across China region were employed by CRU group to generate the CRU time series data (Harris et al., 2014) (above Figure 1), and the WorldClim monthly datasets were generated using all of these stations obtained from the CRU group (Fick and Hijmans, 2017). For independent evaluation of downscaled dataset in this study, these weather stations will be excluded.

We appreciate you very much to point out the issues on the independent evaluation. Now, we have used the 496 independent weather stations to evaluate the original CRU time series (Table 3), WorldClim ( above Tables 1-2), and downscaled time series (Table 3). The results in Table 3 indicated that downscaled datasets had better performance than the original CRU dataset, especially for the 0.5' dataset. We will revised the related contents in the revision.

Table 3. Statistical characteristics between original/downscaled and observed monthly TMPs and PRE in the time series (1951–2016). The values shown here are the averaged evaluation results at all 496 weather stations.

| | Res | MAE$_c$ | MAE$_l$ | MAE$_n$ | RMSE$_c$ | RMSE$_l$ | RMSE$_n$ | NSE$_c$ | NSE$_l$ | NSE$_n$ | Cor$_c$ | Cor$_l$ | Cor$_n$ |
|---|---|---|---|---|---|---|---|---|---|---|---|---|---|
| Minimum | 30' | 1.766 | | | 1.947 | | | 0.887 | | | 0.994 | | |
| TMP (°C) | 10' | 1.673 | 1.515 | 1.558 | 1.802 | 1.726 | 1.793 | 0.896 | 0.902 | 0.899 | 0.995 | 0.995 | 0.995 |
| | 5' | 1.338 | 1.292 | 1.325 | 1.666 | 1.503 | 1.582 | 0.904 | 0.937 | 0.923 | 0.995 | 0.995 | 0.995 |
| | 2.5' | 1.233 | 1.142 | 1.211 | 1.401 | 1.349 | 1.384 | 0.946 | 0.951 | 0.949 | 0.995 | 0.997 | 0.996 |
| | 0.5' | 1.140 | 1.050 | 1.137 | 1.322 | 1.248 | 1.271 | 0.955 | 0.972 | 0.963 | 0.997 | 0.998 | 0.997 |
| Mean | 30' | 1.598 | | | 1.759 | | | 0.888 | | | 0.996 | | |
| TMP (°C) | 10' | 1.277 | 1.140 | 1.188 | 1.433 | 1.293 | 1.358 | 0.899 | 0.914 | 0.904 | 0.997 | 0.997 | 0.997 |
| | 5' | 1.117 | 0.980 | 1.003 | 1.222 | 1.133 | 1.197 | 0.926 | 0.950 | 0.933 | 0.997 | 0.997 | 0.997 |
| | 2.5' | 0.977 | 0.836 | 0.859 | 1.157 | 0.988 | 0.993 | 0.966 | 0.976 | 0.973 | 0.997 | 0.998 | 0.997 |
| | 0.5' | 0.826 | 0.820 | 0.822 | 0.974 | 0.969 | 0.970 | 0.977 | 0.981 | 0.980 | 0.998 | 0.998 | 0.998 |
| Maximum | 30' | 2.034 | | | 2.206 | | | 0.800 | | | 0.995 | | |
| TMP (°C) | 10' | 1.800 | 1.672 | 1.755 | 2.044 | 1.886 | 1.968 | 0.811 | 0.832 | 0.824 | 0.995 | 0.996 | 0.996 |
| | 5' | 1.649 | 1.487 | 1.548 | 1.864 | 1.700 | 1.756 | 0.843 | 0.856 | 0.850 | 0.996 | 0.996 | 0.996 |
| | 2.5' | 1.455 | 1.310 | 1.387 | 1.666 | 1.523 | 1.632 | 0.875 | 0.909 | 0.887 | 0.996 | 0.997 | 0.996 |
| | 0.5' | 1.296 | 1.282 | 1.291 | 1.511 | 1.491 | 1.500 | 0.909 | 0.910 | 0.910 | 0.997 | 0.997 | 0.997 |
| PRE | 30' | 17.850 | | | 29.559 | | | 0.614 | | | 0.885 | | |
| (mm) | 10' | 16.884 | 16.647 | 16.741 | 28.022 | 27.559 | 27.946 | 0.675 | 0.735 | 0.700 | 0.887 | 0.890 | 0.890 |
| | 5' | 16.134 | 15.223 | 15.942 | 26.222 | 25.185 | 25.888 | 0.764 | 0.791 | 0.773 | 0.892 | 0.900 | 0.894 |
| | 2.5' | 14.867 | 14.024 | 14.557 | 24.374 | 23.191 | 23.867 | 0.791 | 0.792 | 0.791 | 0.914 | 0.920 | 0.919 |
| | 0.5' | 13.772 | 13.269 | 13.443 | 22.655 | 21.941 | 22.213 | 0.794 | 0.808 | 0.802 | 0.920 | 0.929 | 0.926 |

Notes: Res indicates the spatial resolution. The subscripts *c*, *l*, and *n* indicate bicubic, bilinear, and nearest-neighbor interpolations, respectively. The original TMPs and PRE are the 30' CRU data and directly compared with the observed data. Evaluations at 10', 5', 2.5', and 0.5' are the evaluations for the downscaled datasets. MAE, RMSE, NSE, and Cor indicate the mean absolute error, root-mean-square error, Nash–Sutcliffe efficiency coefficient, and correlation coefficient.

Specific comments:
1. Figure 1, the range of DEM from 0 to 8848 is wrong. The Turpan Basin is for sure below the mean sea level. What is the spatial resolution of DEM in this figure?
Response: We appreciate you very much to point out this issue. The spatial resolution of DEM in this figure is approximately 1 km. We will revised the Figure 1 in the revision.

2. Figure 2, it is hard to follow this downscaling framework. There is no legend for all figures, which is the mandatory element. The color scales should be the same for a better comparison.
Response: Many thanks for your concerns. We will revised the contents regard to the Delta downscaling framework as well as the schematic illustration. The detailed revision could be found in the responses of General comment #1.

3. More interpretation should be given for the Delta downscaling method. For example, how to calculate the "ratio" for PRE anomaly? Is there a simple mathematical formula?
Response: Many thanks for your suggestion. We will revised these issues in the revision. The detailed revision could be found in the responses of General comment #1.

4. Once again, "Direct evaluation" is not sufficient. More details about the bias or errors should be supplemented.

Response: Many thanks for your comment and suggestion. In the revision, we will revise the "Direct evaluation" and add more details about the bias based on the independent and representative stations. Especially, the spatial distributions of bias at the station scale of original and downscaled CRU time series will be mapped; further, in the geographic space, where have much improvement and what result in the improvement will be analyzed and discussed.